# The E3 ubiquitin ligase IDOL regulates synaptic ApoER2 levels and is important for plasticity and learning

Jie Gao[1†], Mate Marosi[2†], Jinkuk Choi[1], Jennifer M Achiro[3], Sangmok Kim[3], Sandy Li[1], Klara Otis[3], Kelsey C Martin[3], Carlos Portera-Cailliau[2], Peter Tontonoz[1,4,5]*

[1]Department of Pathology and Laboratory Medicine, David Geffen School of Medicine, University of California, Los Angeles, United States; [2]Departments of Neurology and Neurobiology, David Geffen School of Medicine, University of California, Los Angeles, United States; [3]Department of Biological Chemistry, David Geffen School of Medicine, University of California, Los Angeles, United States; [4]Molecular Biology Institute, David Geffen School of Medicine, University of California, Los Angeles, United States; [5]Howard Hughes Medical Institute, David Geffen School of Medicine, University of California, Los Angeles, United States

**Abstract** Neuronal ApoE receptors are linked to learning and memory, but the pathways governing their abundance, and the mechanisms by which they affect the function of neural circuits are incompletely understood. Here we demonstrate that the E3 ubiquitin ligase IDOL determines synaptic ApoER2 protein levels in response to neuronal activation and regulates dendritic spine morphogenesis and plasticity. IDOL-dependent changes in ApoER2 abundance modulate dendritic filopodia initiation and synapse maturation. Loss of IDOL in neurons results in constitutive overexpression of ApoER2 and is associated with impaired activity-dependent structural remodeling of spines and defective LTP in primary neuron cultures and hippocampal slices. IDOL-deficient mice show profound impairment in experience-dependent reorganization of synaptic circuits in the barrel cortex, as well as diminished spatial and associative learning. These results identify control of lipoprotein receptor abundance by IDOL as a post-transcriptional mechanism underlying the structural and functional plasticity of synapses and neural circuits.
DOI: https://doi.org/10.7554/eLife.29178.001

*For correspondence:
ptontonoz@mednet.ucla.edu

†These authors contributed equally to this work

## Introduction

Learning and memory are associated with long-term structural changes at synapses (*Lamprecht and LeDoux, 2004*), and there is a direct relationship between spine morphology and synaptic function (*Dillon and Goda, 2005*). Spines with large heads express higher levels of AMPA receptors and establish stronger synaptic connections (*Tada and Sheng, 2006*). In addition, mounting evidence indicates that synaptic plasticity and learning are associated with changes in spine morphology and that these morphological changes depend on NMDAR activation (*Gambino et al., 2014*; *Hayashi-Takagi et al., 2015*). LTP-inducing stimuli cause the formation of new spines and the enlargement of existing ones (*Matsuzaki et al., 2004*; *Nägerl et al., 2004*), whereas LTD is associated with the shrinkage and elimination of spines (*Okamoto et al., 2004*). Thus, spine morphogenesis and remodeling provide a physical basis for memory in neural networks (*Kasai et al., 2003*). However, the mechanisms that link spine remodeling and synaptic activity remain to be fully elucidated.

ApoE receptors are members of the low-density lipoprotein receptor gene family that play important roles in maintaining the proper function of neurons (*Herz and Bock, 2002*). ApoE Receptor 2

(ApoER2/LRP8) and the very low-density lipoprotein receptor (VLDLR) bind Reelin in addition to ApoE, and ApoER2/VLDLR-mediated Reelin signaling is essential for controlling neuronal migration and positioning during early brain development (*Herz and Chen, 2006*). ApoER2 is also required for the maintenance of proper synaptic function in adulthood (*D'Arcangelo, 2005*). Mice lacking ApoER2 display a defect in LTP while exhibiting normal baseline synaptic transmission (*Weeber et al., 2002*), suggesting a role for ApoER2 in synaptic plasticity.

Given the strong links between ApoER2 and neuronal function, it is reasonable to hypothesize that ApoER2 levels might be coupled to synaptic activity. However, the pathways that control ApoER2 expression and activity in the brain, and the mechanisms by which ApoER2 affects synaptic plasticity are incompletely understood. Long-term synaptic plasticity requires activity-dependent modifications of synaptic strength and connections. This spatially and temporally coordinated process involves multiple components, including changes in glutamate receptor trafficking, cytoskeleton remodeling, and proteasome-mediated local protein degradation (*Mabb and Ehlers, 2010*; *Murakoshi and Yasuda, 2012*; *Shepherd and Huganir, 2007*). The downstream impact of ApoER2 signaling on these components remains to be fully elucidated.

One candidate physiological regulator of ApoER2 signaling is Inducible Degrader of the LDL receptor (IDOL; encoded by the *Mylip* gene), an E3 ubiquitin ligase whose expression is transcriptionally controlled by liver X receptors in (LXRs) in cell types such as hepatocytes and macrophages (*Zelcer et al., 2009*). The IDOL-UBE2D complex ubiquitinates the LDLR on its cytoplasmic domain, thereby targeting it for lysosomal degradation (*Zhang et al., 2011*). Prior studies have shown that the LXR-IDOL pathway regulates cellular cholesterol levels and impacts plasma LDL levels in primates (*Hong et al., 2014*; *Scotti et al., 2011*). Studies in cultured cells have shown that IDOL also promotes the degradation of ApoER2 and VLDLR (*Hong et al., 2010*), but the ability of IDOL to affect the abundance of these lipoprotein receptors in vivo, and the physiological significance of such regulation is unknown. Furthermore, since IDOL and LXR signaling regulate lipoprotein receptors in a cell type-specific manner (*Hong et al., 2014*), the potential impact of IDOL and LXR on neuronal ApoE receptor biology and neuronal physiology cannot yet be predicted.

Here we show that ApoER2 protein levels are actively regulated at the post-translational level in response to neuronal activation and during LTP, and we identify the E3 ubiquitin ligase IDOL (*Zelcer et al., 2009*) as the critical gatekeeper that controls synaptic ApoER2 abundance. IDOL-dependent regulation of ApoER2 dictates dendritic spinogenesis and morphogenesis. Moreover, loss of IDOL impairs activity-dependent remodeling of the synaptic actin skeleton, and compromises synaptic plasticity and cognitive function. These findings reveal a novel post-transcriptional mechanism modulating spine morphogenesis and remodeling in response to synaptic activity.

## Results

### Dynamic regulation of neuronal ApoER2 expression by IDOL

To explore the physiological function of IDOL in neurons, we examined its expression pattern in the brain. We previously generated mice in which the *Mylip* gene was replaced by a knockout cassette containing a LacZ reporter (*Hong et al., 2014*). β-galactosidase staining of sagittal brain sections from IDOL-deficient mice revealed that IDOL was widely expressed in neurons and was particularly abundant in the CA1 region of the hippocampus (*Figure 1A*; *Figure 1—figure supplement 1A and B*). Co-immunofluorescence staining of β-Gal and NeuN (a neuron-specific nuclear protein) in the brain sections of *Mylip*[+/−] mice further confirmed that IDOL was highly expressed in neurons (*Figure 1—figure supplement 1D*).

ApoER2 has been reported to be concentrated in the P2 cellular fraction, which includes postsynaptic densities (*Beffert et al., 2005*). We observed striking increases in VLDLR and ApoER2 protein abundance in total cortical lysates and P2 fractions from IDOL-deficient compared to wild-type (WT) mice (*Figure 1B*). We also found that the induction of mRNA encoding IDOL expression during postnatal mouse brain development between P2 and P22 correlated with decreasing ApoER2 protein despite constant *Lrp8* mRNA expression (*Figure 1C and D*). Moreover, this developmental ApoER2 regulation was abolished in the absence of IDOL (*Figure 1D* lower panel). The increase of IDOL expression in the post-natal period paralleled that of PSD95 and the NR1 subunit of the NMDA

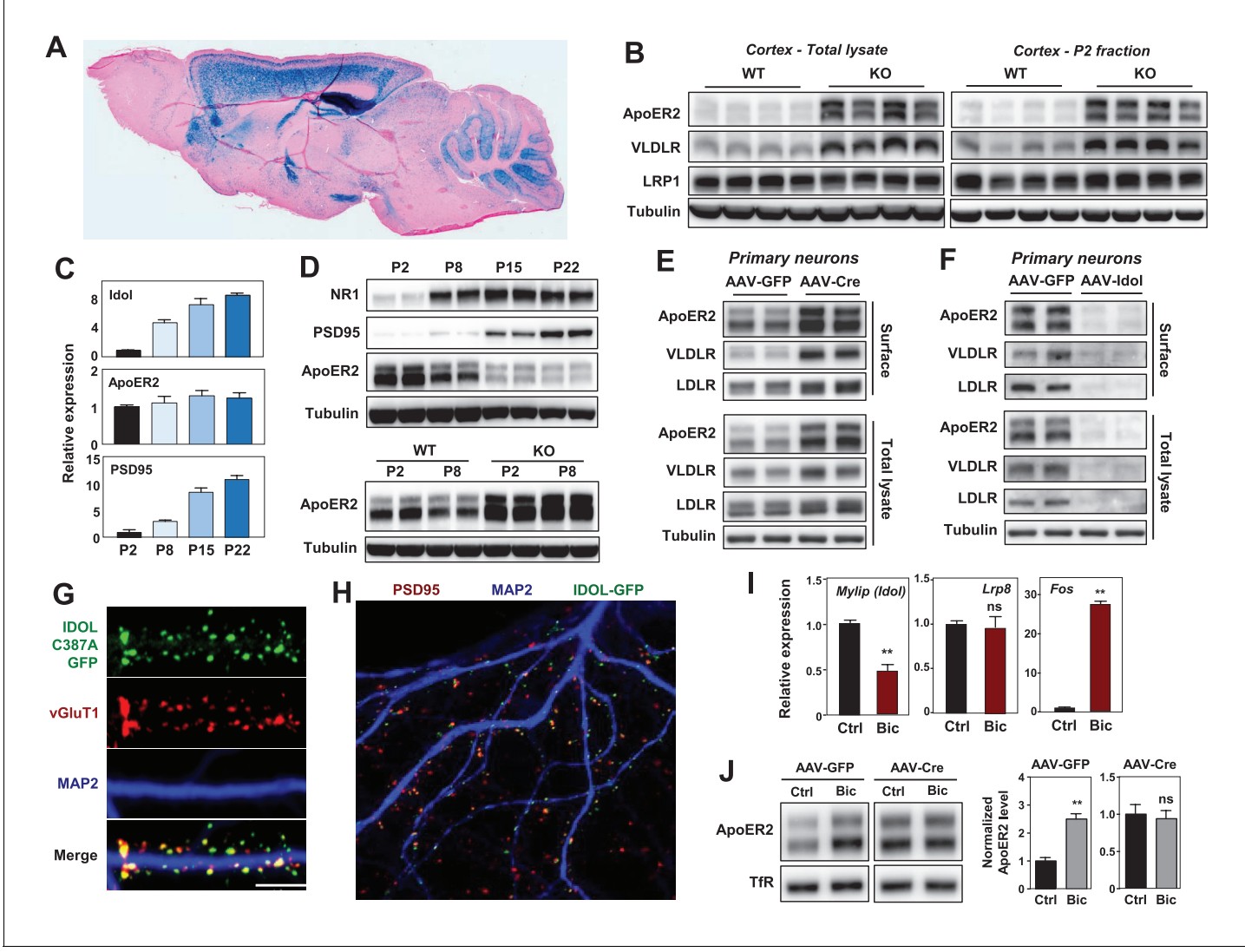

**Figure 1.** IDOL is a dominant regulator of neuronal ApoE receptor abundance. (**A**) Representative image of X-gal staining of sagittal brain sections from 8 week-old IDOL-deficient mice. (**B**) Immunoblot analysis of total protein lysates (left) and crude synaptic fractions (P2) from cortex of 4 week old wild type and IDOL-deficient mice. Lanes represent samples from individual mice. (**C**) Real-time PCR analysis of mRNA expression in cortex of WT mice at postnatal day 2, 8, 15, 22. Data points are means ±S.E.M. (**D**) Immunoblot analysis of total protein lysates from cortex of WT or IDOL-deficient mice at postnatal day 2, 8, 15, 22. Lanes represent samples from individual mice. (**E**) Immunoblot analysis of total protein and biotin-labeled surface protein from *Mylip*flox/flox hippocampal neurons. Neurons were transduced with either AAV-CamKII-GFP or AAV-CamKII-CRE-GFP at DIV 5 and harvested at DIV18-20. Representative data are presented from ≥3 independent experiments. (**F**) Immunoblot analysis of total protein and biotin-labeled surface protein from WT hippocampal neurons. Neurons were transduced with either AAV-hSyn-GFP or AAV-hSyn-IDOL-GFP at DIV 5 and harvested at DIV18-20. Representative data are presented from ≥3 independent experiments. (**G**) Representative images showing immunofluorescence staining of WT hippocampal neurons transfected with a plasmid expressing a GFP-IDOL (C387A) fusion protein. Neurons were transfected at DIV16, and stained and imaged at DIV18. Green, GFP-IDOL (C387A); red, vesicular glutamate transporter 1 (vGluT1); blue, Microtubule-associated protein 2 (MAP2). Scale bars, 10 μm. Representative images are presented from 2 independent experiments, and 6–10 randomly selected neurons are imaged for each experiment. (**H**) Representative images showing immunofluorescence staining of WT hippocampal neurons transfected with a plasmid expressing GFP-IDOL fusion protein. Neurons were transfected at DIV16, and stained and imaged at DIV18. MG132 (10 μM) was added to the culture medium 2 hr before fixation. Green, GFP-IDOL; red, PSD95; blue, Microtubule-associated protein 2 (MAP2). Representative images are presented from 2 independent experiments, and 6–10 randomly selected neurons are imaged for each experiment. (**I**) Real-time PCR analysis of mRNA expression in WT cortical neurons 2 hr after bicuculline (Bic) (40 μM) treatment. Error bars represent SEM. **p<0.01 by Student's t test. (**J**) Immunoblot analysis of biotin-labeled surface protein from *IDOL*flox/flox hippocampal neurons transduced with either AAV-CamKII-GFP or AAV-CamKII-CRE-GFP at DIV 5. Neurons (DIV18) were treated with bicuculline (40 μM) for 2 hr before harvest. Representative data are presented from ≥3 independent experiments. Quantification of ApoER2 levels are shown on the right. Error bars represent SEM. **p<0.01 by Student's t test.

DOI: https://doi.org/10.7554/eLife.29178.002

*Figure 1 continued on next page*

*Figure 1 continued*

The following figure supplements are available for figure 1:

**Figure supplement 1.** IDOL is a dominant regulator of neuronal ApoE receptor abundance.
DOI: https://doi.org/10.7554/eLife.29178.003

**Figure supplement 2.** IDOL is a dominant regulator of neuronal ApoE receptor abundance.
DOI: https://doi.org/10.7554/eLife.29178.004

receptor (*Figure 1D* upper panel), which rise due to the robust synaptogenesis that occurs during this time (*Sans et al., 2000*).

We also found that IDOL bidirectionally regulated ApoER2 and VLDLR protein levels in primary neurons. Deletion of IDOL in *Mylip*$^{flox/flox}$ neurons by means of an AAV-CamKII-*Cre* vector resulted in >95% reduction in mRNA encoding IDOL and strong increases in ApoER2, VLDLR and LDLR protein compared to cells transduced with GFP-expressing AAV vector (*Figure 1E*, *Figure 1—figure supplement 1C*). Conversely, AAV-mediated overexpression of IDOL in primary cortical neurons greatly reduced the levels of these lipoprotein receptors (*Figure 1F*).

There are no antibodies capable of detecting endogenous IDOL protein due to its autocatalytic proteosomal turnover (*Zelcer et al., 2009*). To examine the subcellular localization of IDOL, we transfected a GFP-tagged, catalytically-inactive IDOL fusion protein (GFP-IDOL C387A) (*Scotti et al., 2013*) into primary hippocampal neurons. In immature neurons (7 days of in vitro culture; DIV7), GFP-IDOL distributed widely in soma and dendritic shafts (*Figure 1—figure supplement 2A*). However, in mature DIV18 neurons, GFP-IDOL concentrated at synapses, and was juxtaposed with the glutamatergic presynaptic marker vGlut1 (*Figure 1G*, *Figure 1—figure supplement 2A*). This synaptic enrichment was further confirmed by colocalization of PSD95 and transfected WT GFP-IDOL fusion protein in the presence of MG132 (to block proteosomal degradation of IDOL; *Figure 1H*).

To investigate whether the IDOL-ApoER2 pathway was linked to synaptic activity, we incubated DIV16-18 cortical neurons with the GABAA receptor antagonist bicuculline (Bic), which drives glutamatergic activity. Induction of *Fos*, an immediate early gene, served as a positive control for activation (*Figure 1I*). Bic incubation suppressed *Mylip* mRNA level after 2 hr, and this was associated with an increase in ApoER2 protein, but no change in its mRNA (*Figure 1I,J*). Moreover, Bic-dependent changes in ApoER2 were abolished in neurons lacking IDOL (*Figure 1J*). Similar results were also obtained using KCl to depolarize neurons (*Figure 1—figure supplement 2B*). These data identify IDOL as a dominant regulator of synaptic ApoE receptors under physiological conditions, and show that IDOL is required for the activity-dependent regulation of ApoER2 abundance.

## Excess IDOL activity inhibits spinogenesis by limiting ApoER2 protein

The early postnatal stage is a critical period of spine elaboration, development and maturation (*Yuste and Bonhoeffer, 2004*). The IDOL-dependent regulation ApoER2 at synapses during this period led us to test the involvement of IDOL in spinogenesis. Primary hippocampal neurons were transduced with AAV-hSyn-GFP or AAV-hSyn-GFP-IDOL vectors at DIV 5–7, and then imaged at DIV 18–20. IDOL expression resulted in a marked reduction in spine density (*Figure 2A and B*) and reduced the total and surface levels of key post-synaptic proteins including AMPAR, NMDAR1 and PSD95 (*Figure 2C*). Importantly, the E3 ligase activity of IDOL was required for its effects on synaptogenesis, as expression of an inactive mutant (C387A)(*Zelcer et al., 2009*) failed to recapitulate these effects (*Figure 2C*).

We next investigated whether IDOL regulated spine formation and/or the maintenance of preexisting spines. Primary hippocampal neurons were transduced at DIV 5 and imaged at DIV 9–10, a stage at which most spines are immature and appear as long, thin filopodia. Early IDOL overexpression markedly reduced filopodia density in DIV9-10 neurons (*Figure 2D*). The IDOL C387A mutant had no impact on filopodia density, indicating that E3 ligase activity was required (*Figure 2D*). On the other hand, primary hippocampal neurons that overexpressed IDOL after the major stage of spine maturation (transduced at DIV17 and imaged at DIV 21–22) grossly did not show differences in spine density (*Figure 2E*), and had comparable expression of post-synaptic proteins including AMPAR, NMDAR1 and PSD95 (*Figure 2F*), suggesting that excess IDOL activity affects spine formation but not maintenance.

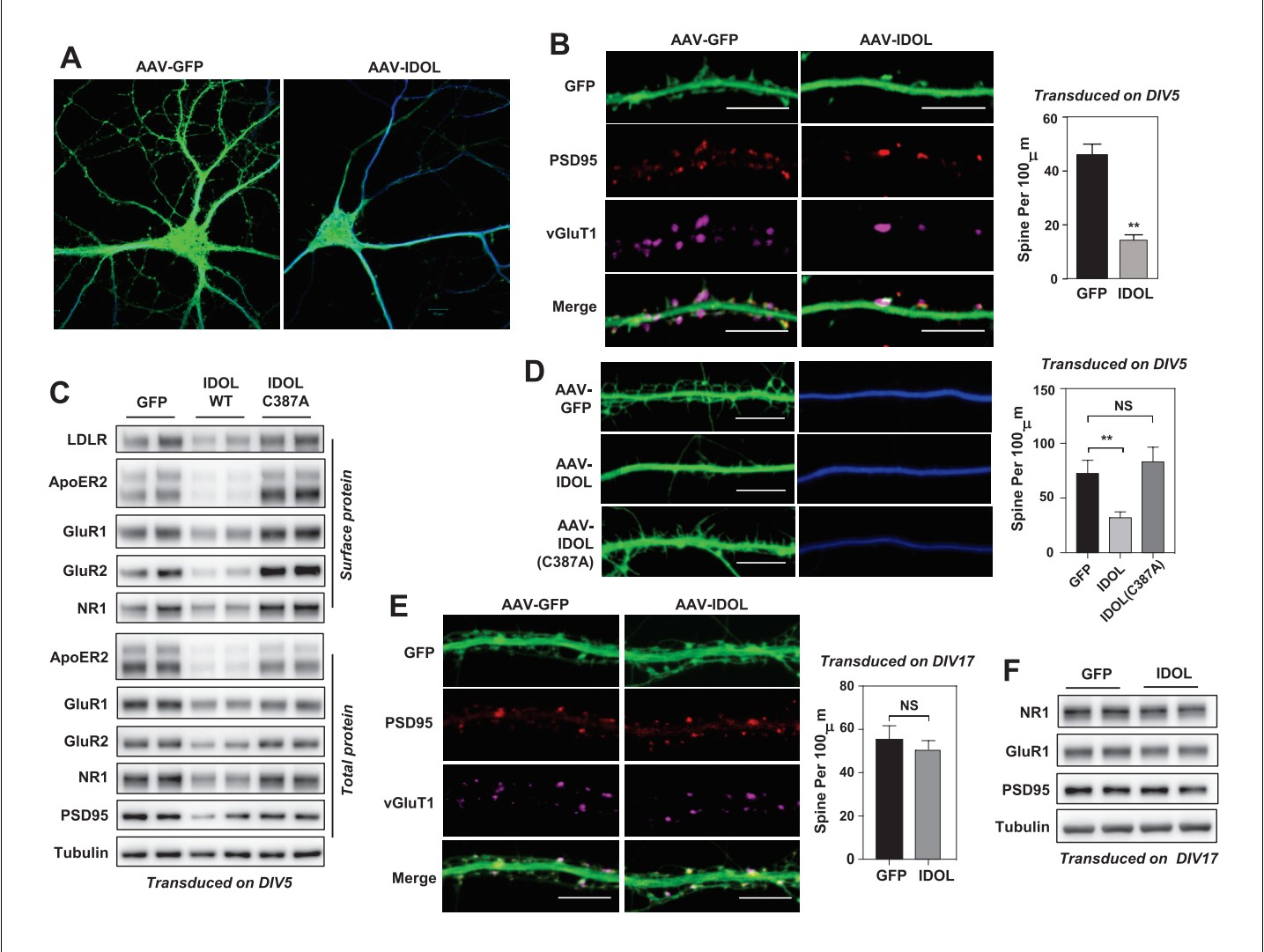

**Figure 2.** Excess IDOL activity inhibits the initiation of dendritic spinogenesis. (**A**) Representative images showing morphology of WT hippocampal neurons transduced with either AAV-hSyn-GFP or AAV-hSyn-IDOL-GFP. Neurons were infected at DIV 5, and fixed and imaged at DIV 20. Images are presented as merged green (GFP) and blue (MAP2) channels. (**B**) Representative images showing morphology of dendrites from WT hippocampal neurons transduced with either AAV-hSyn-GFP or AAV-hSyn-IDOL-GFP. Neurons were infected at DIV 5, and fixed and imaged at DIV 20. Green, GFP; red, PSD95; magenta, vGluT1. Scale bars, 10 μm. Quantification of spine density is shown on the right (n = 8–10/group). **p<0.01 by Student's t test. (**C**) Immunoblot analysis of total and surface protein from WT hippocampal neurons transduced with either AAV-hSyn-GFP, AAV-hSyn-IDOL, or AAV-hSyn-IDOL (C387A). Neurons were infected at DIV 5 and harvested at DIV 20. (**D**) Representative images showing morphology of dendrites from WT hippocampal neurons transduced with either AAV-hSyn-GFP, AAV-hSyn-IDOL-GFP or AAV-hSyn-IDOL (C387A). Neurons were infected at DIV 5, and fixed and imaged at DIV 9. Scale bars, 10 μm. Quantification of spine density is shown on the right (n = 8–10/group). **p<0.01 by Student's t test. (**E**) Representative images showing morphology of dendrites from WT hippocampal neurons transduced with either AAV-hSyn-GFP or AAV-hSyn-IDOL-GFP at DIV 17, and fixed and imaged at DIV 21. Green, GFP; magenta, vGluT1; mlue, MAP2. Scale bars, 10 μm. Quantification of spine density is shown on the right (n = 8–10/group). (**F**) Immunoblot analysis of total protein from WT hippocampal neurons transduced with either AAV-hSyn-GFP or AAV-hSyn-IDOL-GFP at DIV 17, and fixed and imaged at DIV 21.

DOI: https://doi.org/10.7554/eLife.29178.005

To test whether the reduced total and surface levels of key post-synaptic proteins when overexpressing IDOL at early stage of synaptogenesis would lead to decreased basal synaptic transmission, we performed whole-cell patch-clamp recordings of hippocampal neurons transduced with AAV-GFP or AAV-IDOL vectors at DIV 5–7. The recordings of miniature excitatory postsynaptic currents (mEPSCs) at DIV 16–18 showed that IDOL expression reduced their amplitude of but had negligible effects on their frequency (*Figure 3A*), indicating a postsynaptic effect. The knockdown of ApoER2

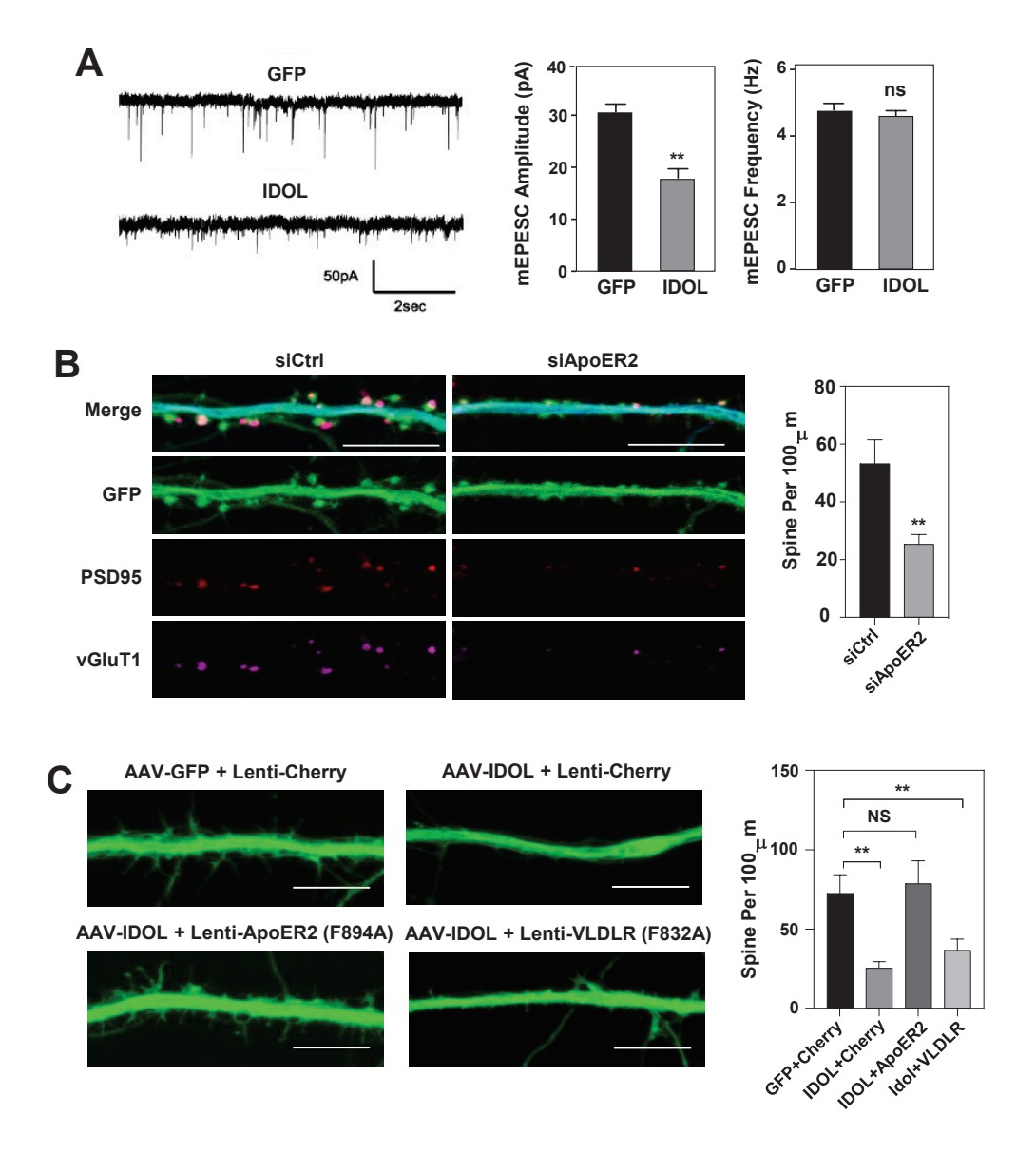

**Figure 3.** IDOL suppresses dendritic spinogenesis through control of ApoER2. (**A**) mEPSC recordings from WT hippocampal neurons transduced with either AAV-hSyn-GFP, or AAV-hSyn-IDOL (right). Neurons were transduced at DIV 5 and recorded between DIV16-18. Quantification of mEPSC amplitude and frequency (n = 7–11) is shown on the right. Error bars represent SEM. **p<0.01 by Student's t test. ns, not significant. (**B**) Representative images showing morphology of dendrites from WT hippocampal neurons incubated with control siRNA (1 μM) or ApoER2 siRNA (1 μM) from DIV5 to DIV16. Green, GFP; red, PSD95; magenta, vGluT1. Scale bars, 10 μm. Quantification of spine density is shown on the right (n = 9–12/group). **p<0.01 by Student's t test. (**C**) Representative images showing morphology of dendrites from WT hippocampal neurons transduced with Lenti-Cherry, Lenti-ApoER2 or Lenti-VLDLR, as well as AAV-hSyn-GFP or AAV-hSyn-IDOL-GFP as labeled. Neurons were infected at DIV 5, and fixed and imaged at DIV 10. Scale bars, 10 μm. All data presented in *Figure 2* are from ≥2 independent experiments. 5–8 randomly selected neurons are imaged for each experiment and used for statistical analysis. Quantification of spine density is shown on the right. **p<0.01 by Student's t test.

DOI: https://doi.org/10.7554/eLife.29178.006

The following figure supplement is available for figure 3:

**Figure supplement 1.** Excess IDOL activity inhibits spinogenesis by limiting ApoER2 protein.

DOI: https://doi.org/10.7554/eLife.29178.007

using siRNA at DIV5 largely recapitulated the effects of IDOL on synaptogenesis (**Figure 3B**). On the other hand, co-expression of an IDOL-resistant ApoER2 mutant (F894A) (**Calkin et al., 2011**) rescued the reduction in filopodia density induced by IDOL overexpression, while an IDOL-resistant VLDLR mutant (F832A) (**Calkin et al., 2011**) failed to do so (**Figure 3C**). In addition, the ability of IDOL to reduce the levels of key post-synaptic proteins were preserved in $Ldlr^{-/-}$ neurons (**Figure 3—figure supplement 1A**). And adding neither RAP (a high-affinity competitive ligand for ApoE receptors) nor neutralizing antibody to Reelin (CR50) abolished the effects of IDOL on the expression of key post-synaptic proteins (**Figure 3—figure supplement 1B**). Together, these data suggested that the inhibitory effects of IDOL on spinogenesis were likely mediated through ApoER2, and largely independent of the Reelin signaling pathway and the uptake of lipoprotein particles in our experimental setting.

## IDOL-mediated ApoER2 regulation is essential for spine morphogenesis

To better understand the importance of physiological IDOL expression in spinogenesis, we transduced primary hippocampal neurons from $Mylip^{flox/flox}$ mice with CamKII-driven AAV vectors expressing GFP or GFP-Cre at DIV 5–7. Mature spines are characterized by bulbous heads that are connected to the dendritic shafts by narrow necks (**Hering and Sheng, 2001**). At DIV18-21, control neurons showed numerous mature dendritic spines with typical morphology (**Figure 4A**, **Figure 4—figure supplement 1A**). By contrast, IDOL-deficient neurons displayed many filopodia-like, thin, immature spines with no enlarged spine head. Spine density was increased by 42%, while spine head width decreased approximately 50% (**Figure 4A**). Immunostaining of IDOL-deficient neurons with the postsynaptic marker PSD95 and presynaptic marker vGluT1 further revealed that the tips of many dendritic protrusions were not apposed to presynaptic terminals (**Figure 4A**). In contrast to control neurons, in which most of the synapses were formed at the spine heads (white arrows), IDOL-deficient had numerous excitatory synapses formed directly on the dendritic shafts (red arrows) (**Figure 4A**). These data suggest that IDOL is important for the proper spine morphogenesis during filopodia-to-mature spine transition process.

To test whether ApoER2 mediates the effects of IDOL on spine maturation, we transduced primary hippocampal neurons with AAV-GFP and lentivirus vectors expressing mCherry or ApoER2. ApoER2 overexpression resulted in a 2-fold increase in spine density and a ~50% reduction in head width, largely recapitulating the effects of IDOL deficiency (**Figure 4B**). We further reasoned that if excessive ApoER2 expression was the cause of the spine phenotype, then knocking down ApoER2 in IDOL-deficient neurons should normalize morphology. Indeed siRNA-mediated ApoER2 knockdown increased the width of spine heads in IDOL-deficient hippocampal neurons (**Figure 4C**, **Figure 4—figure supplement 1B**). Thus, precise control of synaptic ApoER2 protein levels by IDOL is critical for spine morphogenesis. Either too little or too much ApoER2 leads to dysmorphic effects.

To assess the relevance of these findings in vivo, we employed Golgi-impregnation to visualize the morphology of CA1 pyramidal neurons from 5 week-old mice. Spine images were taken at a distance of ~150 μm from the base of apical dendrites to minimize variation. Consistent with results from primary neuron cultures, IDOL-deficient mice had higher spine density but reduced head width compared with their WT counterparts (**Figure 4D**).

## The IDOL-ApoER2 axis modulates the coupling between glutamate receptors and Rac1

Given the impact of the IDOL-ApoER2 pathway on neuronal morphogenesis, we investigated its effects on signaling downstream of glutamate receptor activation. Deletion of IDOL from hippocampal neurons did not change total or surface levels of the GluR1 and GluR2 subunits of the AMPA (α-amino-3-hydrozy-5-methylisoxa-zole-4-propionic acid) receptor, nor the NR1, NR2a or NR2b subunits of NMDARs (**Figure 5A**). ApoER2 has been reported to interact directly with NR1, but co-immunostaining with vGluT1 revealed that NR1 was appropriately localized on dendritic spines in IDOL-deficient hippocampal neurons (**Figure 5B**). Consistent with these findings, total lysates and P2 fractions from the brain of 4-week-old IDOL-deficient mice showed comparable levels of glutamate receptors to WT mice (**Figure 5C**). However, loss of IDOL was associated with reduced levels of the scaffold protein PSD95 in the P2 fraction, possibly reflecting less scaffolding protein embedded at the postsynaptic density of the smaller spine heads observed in IDOL-deficient (**Figure 5C**).

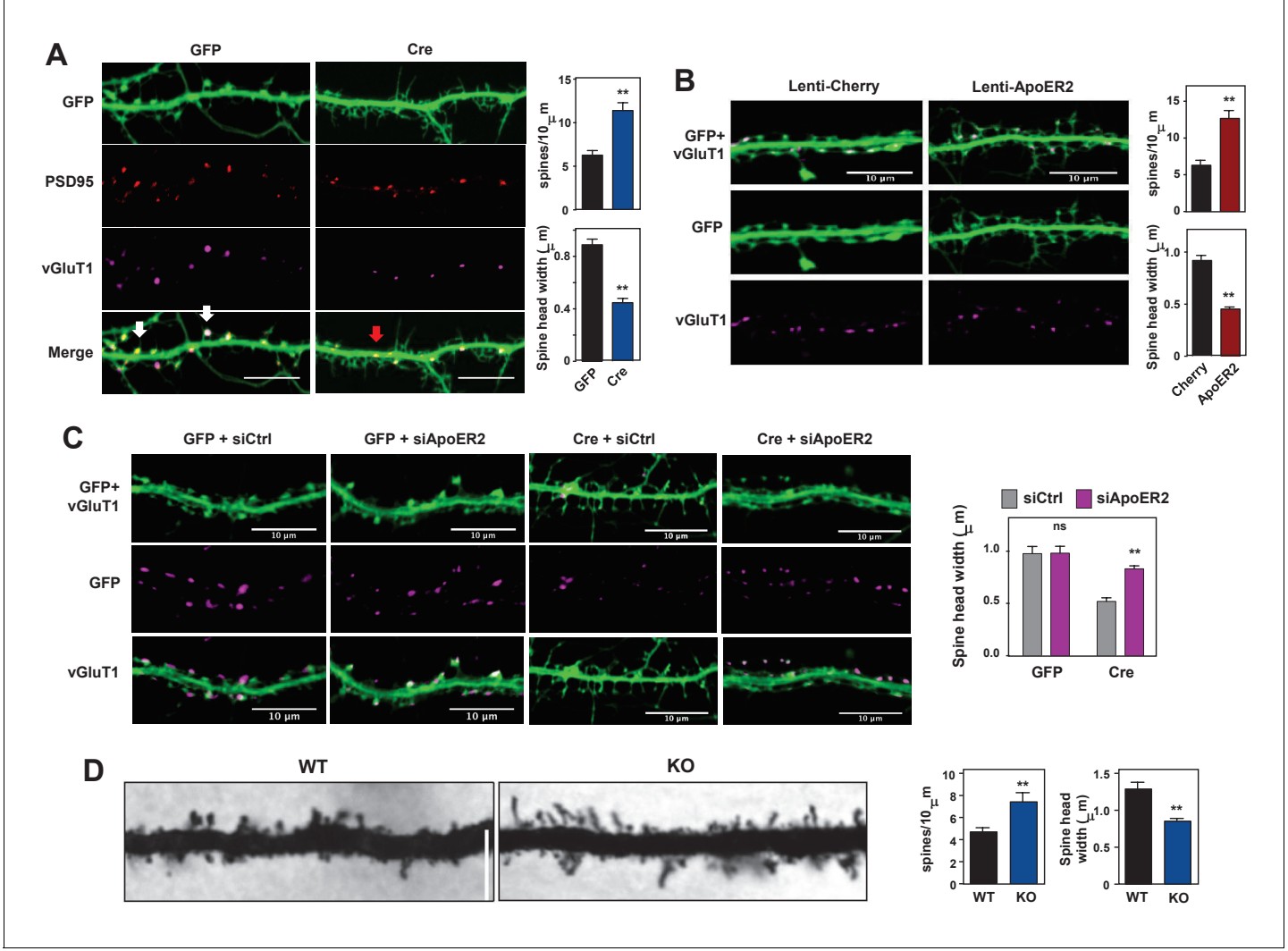

**Figure 4.** Loss of IDOL impairs dendritic Spine morphogenesis and maturation. (**A**) Representative images showing immunofluorescence staining of dendrites from *Mylip*flox/flox hippocampal neurons transduced with either AAV-CamKII-GFP or AAV-CamKII-CRE-GFP. Neurons were transduced at DIV 5, and fixed and stained at DIV 19–20. Spine synapse, white arrow; Shaft synapse, red arrow; Green, GFP; Red, PSD95; Magenta, vGluT1. Scale bars, 10 μm. Quantification of spine density and spine-head width of neurons (including filopodia-like protrusions) is shown on the right (n = 10–12/group). **p<0.01 by Student's t test. (**B**) Representative images showing immunofluorescence staining of dendrites from WT hippocampal neurons transduced with either Lenti-Cherry or Lenti-ApoER2. Neurons were transduced at DIV 10, and fixed and stained at DIV 19–20. Both groups were also transduced with a AAV-CamKII-GFP DIV15 to allow visualization of spines. Green, GFP; magenta, vGluT1. Scale bars, 10 μm. Quantification of spine density and spine-head width of neurons from (including filopodia-like protrusions, n = 10–12/group) is shown on the right. **p<0.01 by Student's t test. (**C**) Representative images showing immunofluorescence staining of dendrites from *Mylip*flox/flox hippocampal neurons transduced with either AAV-CamKII-GFP or AAV-CamKII-CRE-GFP at DIV10. Control siRNA (1 μM) or ApoER2 siRNA (1 μM) were then added to the culture medium at DIV16 and applied again after 48 hr. Neurons were fixed and stained at DIV21. Green, GFP; magenta, vGluT1. Scale bars, 10 μm. Quantification of spine-head width of neurons (including filopodia-like protrusions, n = 10–12/ group) is shown on the right. (**D**) Representative images of primary CA1 apical dendrites (about 150 μM from the soma) in Golgi-impregnated slices from 5-weeks-old WT and IDOL-deficient mice. Scale bars, 5 μm; quantification of dendritic spine density along the dendrites of Golgi-impregnated CA1 hippocampal neurons obtained from WT (n = 9 total neurons, 3 animals) and IDOL-deficient mice (n = 10 total neurons, 3 animals) is shown on the right. Error bars represent SEM. **p<0.01 by Student's t test. Primary neuron images presented in this figure are from ≥2 independent experiments, 5–8 randomly selected neurons are imaged for each experiment and used for statistical analysis.
DOI: https://doi.org/10.7554/eLife.29178.008

The following figure supplement is available for figure 4:

**Figure supplement 1.** Loss of IDOL impairs dendritic Spine morphogenesis and maturation.
DOI: https://doi.org/10.7554/eLife.29178.009

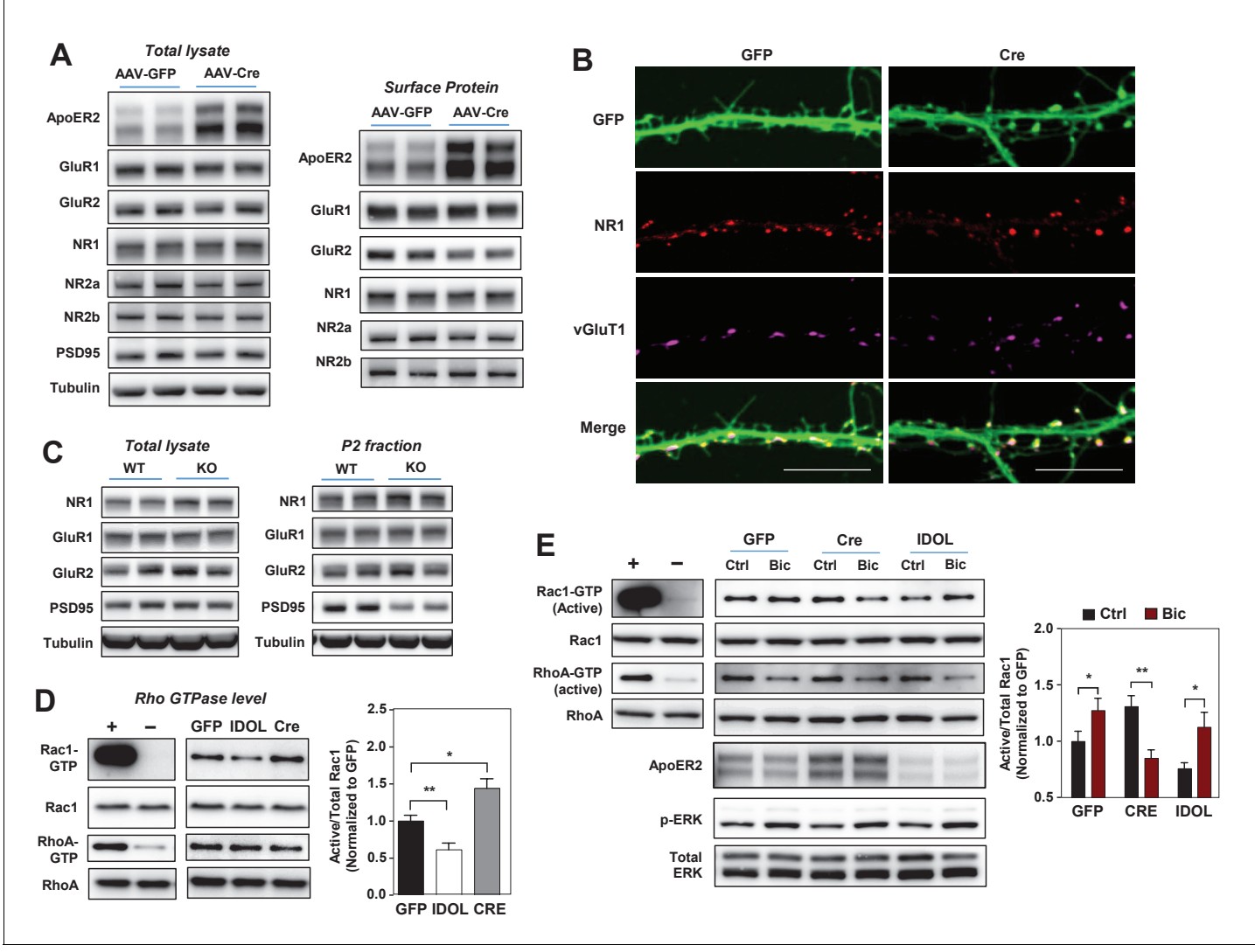

**Figure 5.** The IDOL-ApoER2 axis modulates the coupling between synaptic activity and Rac1. (**A**) Immunoblot analysis of total protein and biotin-labeled surface protein from *Mylip*[flox/flox] hippocampal neurons transduced with either AAV-CamKII-GFP or AAV-CamKII-CRE-GFP. Neurons were transduced at DIV 5 and harvested at DIV 18. Each lane represents pooled samples from two wells of a 12-well-plate. Representative data are presented from ≥3 independent experiments. (**B**) Representative images showing immunofluorescence staining of dendrites from *Mylip*[flox/flox] hippocampal neurons transduced with either AAV-CamKII-GFP or AAV-CamKII-CRE-GFP. Neurons were transduced at DIV 5, and fixed and stained at DIV 20. Green, GFP; red, NR1. Scale bars, 10 μm. (**C**) Immunoblot analysis of total protein lysates (left) and crude synaptic fractions (P2) from cortex of 4-week-old WT and *IDOL*[−/−] mice. Lanes represent samples from individual mice. Representative data are presented from samples of four individual mice per group. (**D**) Immunoblot analysis of endogenous GTP-bound (active) Rac1 and Rho, and total Rac1 and Rho in the protein lysate from *Mylip*[flox/flox] cortical neurons transduced with AAV vectors expressing GFP, CRE or IDOL. Neurons were transduced at DIV 5 and harvested at DIV 16–18. Each lane represents pooled samples from three wells of a 12-well-plate. Representative data are presented from ≥2 independent experiments. Quantification of endogenous GTP-bound (active) Rac1 levels is shown on the right. *p<0.05 **p<0.01 by Student's t test. (**E**) Immunoblot analysis of endogenous GTP-bound (active) Rac1 and Rho, and total Rac1 and Rho in the protein lysates from *Mylip*[flox/flox] cortical neurons transduced with either AAV-CamKII-GFP, AAV-CamKII-CRE-GFP or AAV-CamKII-IDOL-GFP. Neurons were transduced at DIV 5 and harvested at DIV 16–18. Neurons were treated with bicuculline (40 μM) for 10 min before harvest. Each lane represents pooled samples from three wells of a 12-well-plate. Representative data are presented from ≥2 independent experiments. Quantification of endogenous GTP-bound (active) Rac1 in the protein lysates is shown on the right. *p<0.05 **p<0.01 by Student's t test. Primary neuron images presented in this figure are from ≥2 independent experiments, 5–8 randomly selected neurons are imaged for each experiment and used for statistical analysis.

DOI: https://doi.org/10.7554/eLife.29178.010

The following figure supplement is available for figure 5:

**Figure supplement 1.** IDOL effects on Rac1 activity involve JIP1 and TIAM1.

DOI: https://doi.org/10.7554/eLife.29178.011

Dendritic spine formation and morphogenesis are mainly determined by the dynamics of the actin cytoskeleton. Rho family small GTPases are critical molecular switches controlling actin remodeling during spinogenesis (*Dillon and Goda, 2005*). The reduced spine formation observed with IDOL overexpression, and the increased proportion of filopodia with IDOL knockout, are reminiscent of phenotypes caused by inactivation or constitutive activation of the GTPase Rac1, respectively (*Nakayama et al., 2000*; *Zhang and Macara, 2006*). We found that IDOL overexpression decreased, while IDOL knockout increased, the basal level of GTP-bound (active) Rac1 in primary cortical neurons (*Figure 5D*). The induction of Rac1 activity downstream of glutamate receptors, particularly NMDAR, plays a central role in promoting actin polymerization and enlargement of the spine head (*Xie et al., 2007*). We therefore further tested whether IDOL affects Rac1 activity in response to neuronal activity. In control primary neurons, Bic treatment induced Rac1 activity as expected (*Figure 5E*). Interestingly, although IDOL-deficient neurons showed increased basal Rac1 activity, Bic failed to induce, and actually reduced, Rac1 activity (*Figure 5E*). However, induction of ERK phosphorylation in response to Bic treatment was not impaired by either IDOL knockdown or overexpression (*Figure 5E*). This observation suggests that the IDOL-ApoER2 axis selectively affects signaling pathways related to actin remodeling. Supporting the notion that ApoER2 is the principle mediator of IDOL effects on Rac1, the Rac1 activity phenotype of neurons overexpressing ApoER2 resembled that of IDOL-deficient neurons (*Figure 5—figure supplement 1A*): both showed increased basal Rac1 activity and compromised Rac1 induction upon Bic treatment.

It has previously been reported that c-jun-terminal kinase interacting proteins 1/2 (JIP1/2) interact with ApoER2 (*Stockinger et al., 2000*) as well as Tiam1 (*Buchsbaum et al., 2002*), the major Rac guanine nucleotide exchange factor that couples the NMDAR to Rac1 (*Tolias et al., 2005*). We found that loss of IDOL increased protein levels of JIP1 in both total brain lysates and P2 fractions (*Figure 5—figure supplement 1B*). We speculated that this increased JIP1 l in IDOL-deficient neurons might interfere with the interaction between NMDAR and Tiam1. Consistent with this idea, loss of IDOL did not affect protein levels of Tiam1 (*Figure 5—figure supplement 1C*), but markedly altered its localization in primary neurons (*Figure 5—figure supplement 1D*). In WT neurons Tiam1 was enriched within the spine heads as expected, but it was mislocalized to the base of the spines in IDOL-deficient neurons. These observations suggest that changes in JIP1 and Tiam1 may contribute to the decoupling between NMDAR and Rac1 observed in the absence of IDOL.

## Loss of IDOL impairs LTP in primary neurons and hippocampal slices

Alteration of dendritic spine morphology and density in response to neuronal activity is a vital component of neural plasticity (*Hotulainen and Hoogenraad, 2010*). We therefore hypothesized that IDOL may also impact synaptic plasticity. We first tested the impact of IDOL on glycine-induced LTP in cultured hippocampal neurons. Brief stimulation of postsynaptic NMDAR induces rapid insertion and clustering of AMPA receptors at synapses and consequently enhanced mEPSCs (*Lu et al., 2001*). Consistent with our prior findings that neuronal activity determined ApoER2 protein levels, we found that glycine treatment increased ApoER2 protein in hippocampal neurons after 40 min, and that this effect was entirely dependent on IDOL expression (*Figure 6A*). Basal mEPSCs recordings of hippocampal neurons did not reveal changes in either amplitude or frequency of mEPSCs upon IDOL deletion (*Figure 6B*), suggesting that although loss of IDOL affects spine morphogenesis, it does not have a major detrimental impact on basal synaptic transmission. The amplitude and frequency of mEPSCs were increased in GFP-expressing hippocampal neurons after stimulation with glycine as expected. By contrast, glycine stimulation did not produce increased mEPSCs in IDOL-deficient neurons, suggesting a defect in LTP (*Figure 6B*).

The failure to enhance mEPSCs upon glycine treatment was unlikely to be due to a complete block in NMDAR activation, as glycine treatment induced phosphorylation of ERK and GluR1 (S845) to a similar extent in WT and IDOL-deficient neurons (*Figure 6—figure supplement 1*). Rather, we found that loss of IDOL led to defective structural remodeling during chemical LTP. Glycine treatment caused a 42% increase in spine number in WT neurons (*Figure 6C and D*), and we observed more spines with head width >0.8 µM (*Figure 6D*). By contrast, glycine treatment failed to increase either spine number or head width in IDOL-deficient neurons (*Figure 6D*). These findings suggest that defects in spine structural remodeling upon NMDAR activation may underlie the failure to induce LTP in the absence of IDOL.

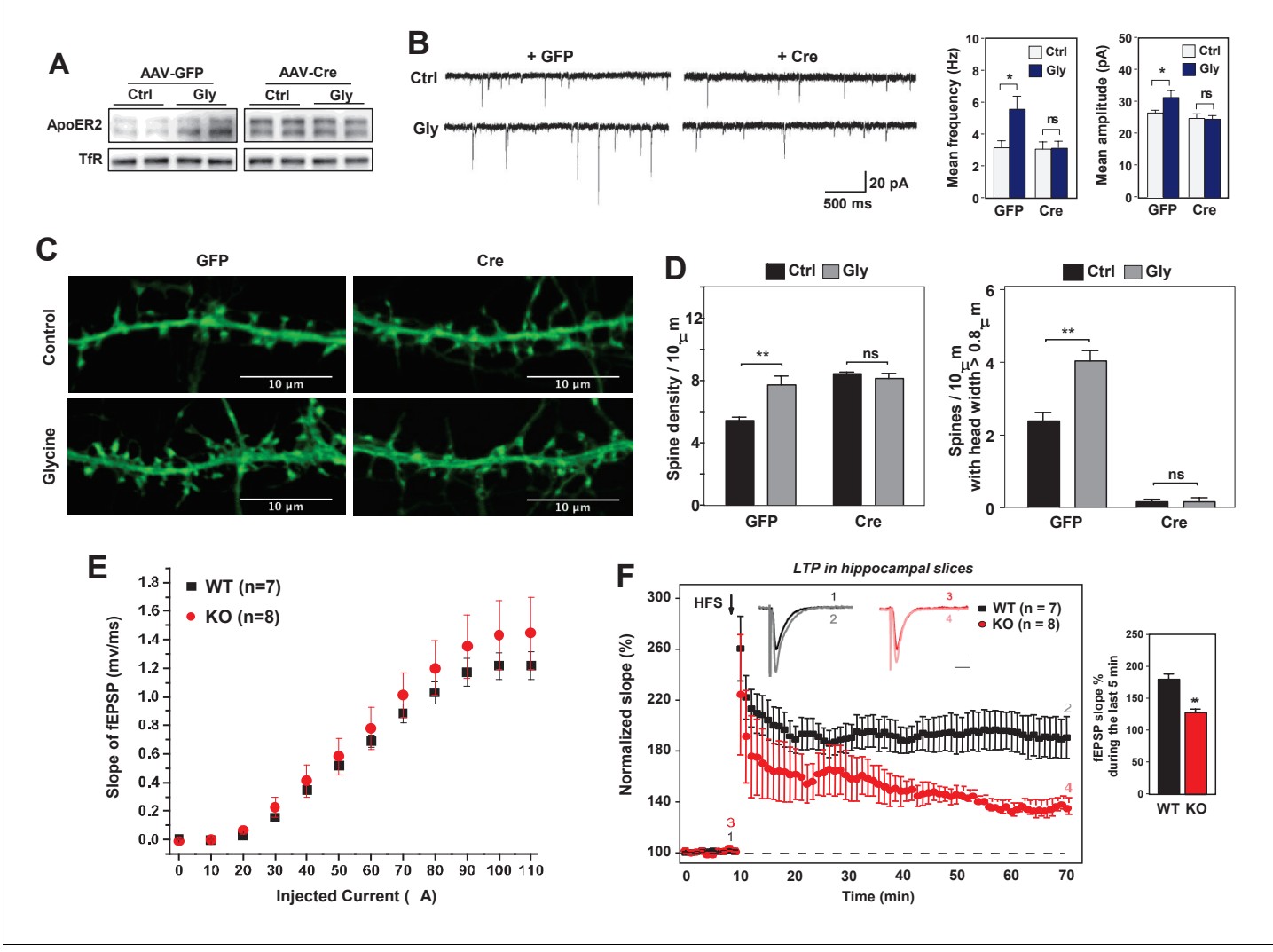

**Figure 6.** Loss of IDOL impairs LTP in neurons and hippocampal slices. (A) Immunoblot analysis of total proteins from *Mylip*^flox/flox hippocampal neurons transduced with either AAV-CamKII-GFP or AAV-CamKII-CRE-GFP at DIV 5. At DIV21, neurons were briefly treated with glycine (200 µM) followed by 30–40 min resting before harvest (cLTP protocol in Materials and Methods). Each lane represents pooled samples from two wells of a 12-well-plate. Representative data are presented from 2 independent experiments. (B) (Left) mEPSC recordings from *Mylip*^flox/flox hippocampal neurons with/without glycine treatment (cLTP protocol). Neurons were transduced with either AAV-CamKII-GFP or AAV-CamKII-CRE-GFP at DIV 5 and recorded between DIV19-21. (Right) Quantification of mEPSC amplitude and frequency (n = 9–15/group). Error bars represent SEM. *p<0.05 by Student's t test. ns, not significant. (C) Representative images showing morphology change of dendritic spines with/without glycine treatment (cLTP protocol). (D) Quantification of spine density and number of spines with big head width (>0.8 µM) of neurons before and after cLTP protocol in *Figure 6C*. n = 10–12/ group. (E) Input–output curves of fEPSP slopes in the WT group (■),IDOL-deficient group (•). Data points are means ±S.E.M. (WT n = 7 and IDOL-deficient n = 8 slices). (F) (Left) LTP induced by HFS (arrow) was reduced in the SC-CA1 synapses of IDOL-deficient mice (•) compared to WT mice (■). Data points are means ±S.E.M. of normalized slopes of fEPSPs in every minute (3 traces/min). Insets show representative traces of evoked EPSPs recorded in hippocampal slices of WT and IDOL-deficient mice before (black, red) and after HFS (gray, light red) respectively. Calibration: 0.2 mV, 10 msec. Numbers (1, 2, 3, 4) show the origin of the representative traces. (Right) Summary of magnitude of LTP (% baseline EPSP slope; means ±S.E.M.) during the last 5 min (between 55 and 60 min) of the recording (n = 7 slices from 6 WT mice and n = 8 slices from 6 IDOL-deficient mice). **p<0.01 by Mann–Whitney U-test.

DOI: https://doi.org/10.7554/eLife.29178.012

The following figure supplement is available for figure 6:

**Figure supplement 1.** Loss of IDOL impairs LTP in neurons and hippocampal slices.

DOI: https://doi.org/10.7554/eLife.29178.013

We next asked whether hippocampal LTP at Schaffer collateral-CA1 pathways was altered in the absence of IDOL. We first examined the basal synaptic properties to evaluate the possible impairment of Schaffer collateral-CA1 synaptic transmission in IDOL-deficient hippocampal slices. Input-Output (IO) curves were established by plotting the fEPSP slope against various intensities of the test pulse. We found no difference in the IO curves between WT and IDOL-deficient slices (*Figure 6E*), suggesting that basal synaptic function was not affected by the absence of IDOL. In both WT and IDOL-deficient slices, a standard protocol of high frequency stimulation (HFS) of Schaeffer collateral synapses caused a robust increase in the slope of the fEPSPs that persisted throughout the 1 hr recording period after HFS (*Figure 6F*). However, whereas the magnitude of LTP remained at a persistently high level in WT slices for the duration of the recording, in IDOL-deficient slices the magnitude of LTP gradually decreased over time such that, by the end of the recording, the potentiation in the slope of the fEPSPs was markedly reduced compared to WT mice (*Figure 6F*; 186.1 ± 6.4% in WT vs. 133.0 ± 2.5% in IDOL KO, p=0.00654, Mann-Whitney U-test).

## IDOL is required for experience-dependent neural plasticity in vivo

To investigate the role of IDOL in experience-dependent plasticity of neuronal circuits in the barrel cortex, we employed longitudinal optical intrinsic signal (OIS) imaging through cranial windows to delineate single whisker response maps in the barrel cortex. The OIS signals were elicited in response to mechanical stimulation of the contralateral D2 whisker (D2W) and hind-limb (HL) – the latter was used as a control – through glass-covered cranial windows (*Figure 7A*, *Figure 7—figure supplement 1A*). Baseline OIS imaging was acquired in 2 separate sessions (1 week apart) and averaged. There was no significant difference in the cortical map size for D2W at baseline between WT (N = 7) and IDOL-deficient (N = 8) groups (*Figure 7—figure supplement 1B*). Immediately after the last baseline imaging session we performed unilateral plucking of all whiskers (down to the follicles) contralateral to the cranial window, with the exception of the D2 whisker. OIS imaging of the D2 whisker representation was performed 7, 14, and 21 days after whisker deprivation.

In WT mice, we observed an increase in the size of the functional representation of the spared D2 whisker relative to the pre-deprivation baseline beginning at 7 days after whisker plucking and persisting throughout the imaging period (*Figure 7B and C*). This degree of plasticity is consistent with published studies showing an expansion of the cortical representation of spared whiskers after sensory deprivation (*Margolis et al., 2012*; *Polley et al., 1999*). By contrast, there was no increase in the area of the functional D2W representation in IDOL-deficient mice at any time after whisker deprivation (*Figure 7B and C*). When comparing the two genotypes, we observed a significantly larger D2W map size in WT mice after whisker plucking (Two-way ANOVA with repeated measures, interaction between genotype and time p=0.004). This difference was most noticeable at +21 d, which coincides with the time when previous studies have detected the largest effect of whisker deprivation on OIS map size (*Margolis et al., 2012*). We found a + 50% change in map size from baseline in WT compared to no change in IDOL-deficient mice (*Figure 7D*). In control experiments, we found that whisker deprivation did not affect the cortical hind limb (HL) map in WT control mice or in IDOL-deficient mice (*Figure 7—figure supplement 1C,D*). These findings demonstrate that IDOL plays an indispensable role in the experience-dependent plasticity of neural circuits in vivo.

## Loss of IDOL impairs learning and memory in mice

We further assessed the importance of the IDOL pathway in hippocampal-dependent spatial and contextual memory. First, we tested control and IDOL-deficient mice in the standard Morris water maze with a hidden platform (*Vorhees and Williams, 2006*). WT mice performed better than IDOL-deficient mice in this task, as measured by reduced escape latencies over the course of the training trials, although both groups improved (*Figure 8A*). To determine the degree of reliance of the mice on spatial versus non-spatial cues to find the platform, we performed probe trials in which the platform was removed. WT mice spent significantly more time in the target quadrant after 7 or 9 days training, while IDOL-deficient mice showed no improvement (*Figure 8B*). This finding suggested that IDOL-deficient mice failed to acquire the memory for the location of the hidden platform. These performance differences could not be attributed to vision or locomotor defects, as both groups performed similarly in the visible platform test (*Figure 8C*), and exhibited comparable swimming speed (*Figure 8D*). We also tested contextual fear conditioning to evaluate hippocampal-dependent fear

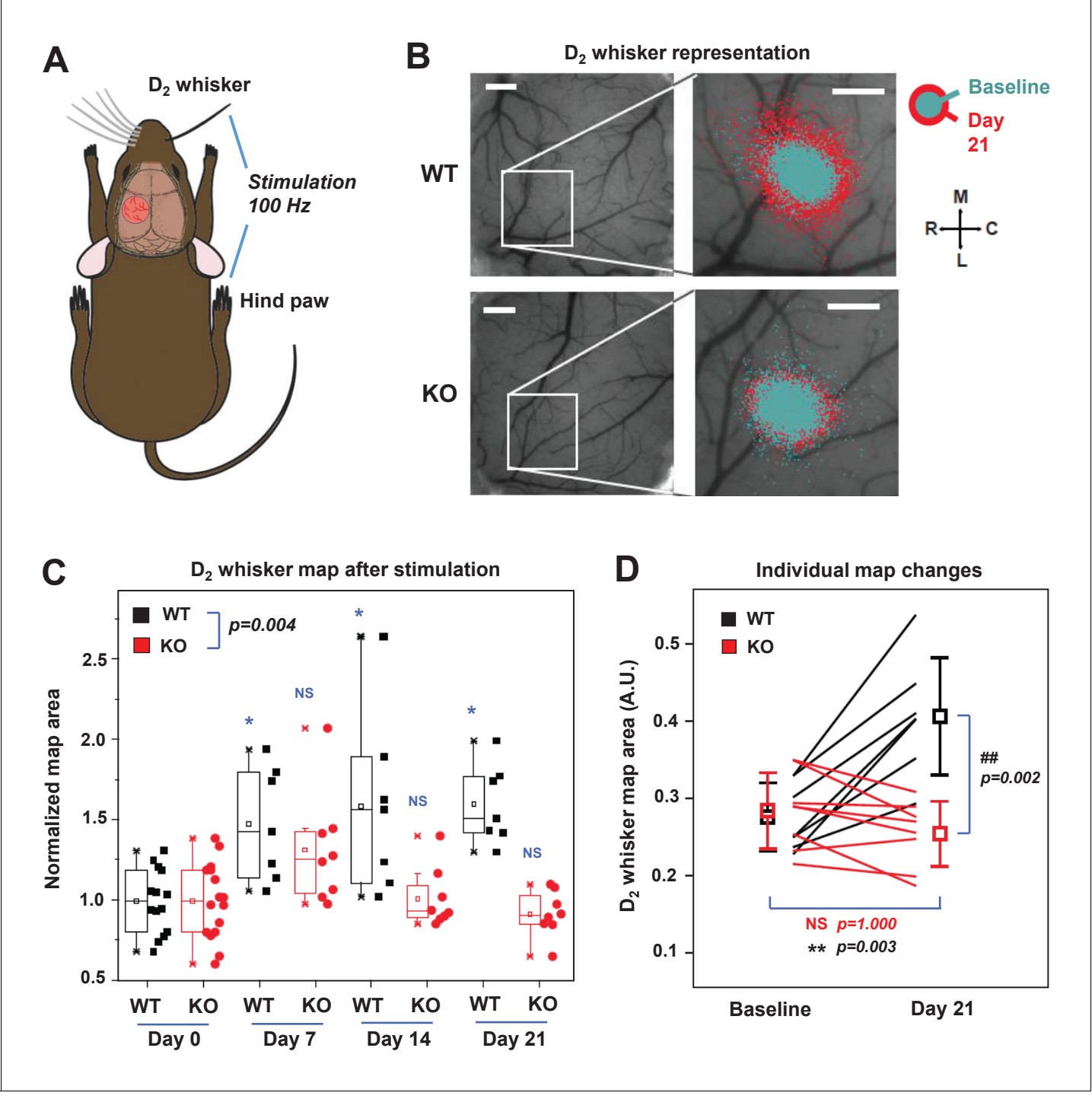

**Figure 7.** IDOL is required for experience-dependent neural plasticity. (**A**) Schematic of experimental design for Optical Intrinsic Signal (OIS) experiments. We mechanically stimulated (piezoelectric square deflections, t = 1.5 s, f = 100 Hz, anterior-posterior deflections) the contralateral D2 whisker (D2W), and hindpaw (HP). (**B**) OIS images showing the functional representation of the D2W map at baseline (cyan) and at +21 d after plucking all whiskers except the D2W (red) for both WT mice (top row) and IDOL-deficient mice (bottom row). Scale bars: 0.5 mm. Orientation: M – medial, L – lateral, R – rostral, C – caudal. (**C**) Box-plot graph (95% CI) showing the differences between normalized sensory map size of D2W at baseline (see Materials and Methods), and at +7 days,+14 days and +21 days after whisker deprivation in WT (n = 7) and knockout (n = 8) mice. Each symbol represents an individual animal. -: median, □: mean. (*p<0.05; **p<0.01 vs. control within genotypes; # p<0.05; ## p<0.01 between experimental groups – Two-way ANOVA with repeated measures followed by Bonferroni's multiple comparison test). There was a significant difference between groups with time as a factor (Two-way ANOVA with repeated measures, interaction between genotype and time p=0.0034). (**D**) Comparison of the average size of the D2W functional representation maps between WT (black) and IDOL-deficient mice (red) at baseline and over 21 days after whisker

*Figure 7 continued on next page*

*Figure 7 continued*

plucking. Lines represent individual mice. □: averages ± S.D. **p=0.003, for the baseline vs. whisker deprivation in WT mice. ##p=0.00199 for comparison between WT and knockout mice.

DOI: https://doi.org/10.7554/eLife.29178.014

The following figure supplement is available for figure 7:

**Figure supplement 1.** IDOL is required for experience-dependent neural plasticity.

DOI: https://doi.org/10.7554/eLife.29178.015

learning (*Curzon et al., 2009*). Both WT and IDOL-deficient mice were trained to associate a novel environment with a foot-shock, and then placed back in the same environment 24 hr later. Freezing was measured over 5 min to assess how well the association was learned. IDOL-deficient mice spent less time freezing than WT mice, indicating a failure to associate the foot-shock with the context (*Figure 8E*).

We also assessed the consequence of excess IDOL expression in the hippocampus, which would be expected to abolish ApoER2 protein expression (see *Figure 1*). Stereotactic injection of AVV-IDOL into the hippocampus of C57Bl/6 mice (*Figure 8F*) was associated with decreased performance in the fear-conditioning test compared to mice injected with AAV-GFP control virus (*Figure 8G*). The finding that excess IDOL expression in hippocampus impairs learning and memory is consistent with prior studies showing that loss of ApoER2 impairs these processes (*Weeber et al., 2002*). The finding that loss of IDOL also impairs learning and memory strongly reinforces the conclusion that precise controls of ApoER2 protein levels are important for normal brain function. Either too little or too much ApoER2 is detrimental.

## Discussion

Our data implicate the IDOL-ApoER2 axis as a key player in regulating dendritic spine initiation and morphogenesis and their structural remodeling downstream of glutamate receptor activation. Loss of IDOL does not globally disrupt CNS development or basal synaptic transmission, but greatly impairs the functional plasticity of synapses and neural circuits in vivo. IDOL-deficient neurons are defective in their ability to undergo activity-dependent spine remodeling and IDOL-deficient mice show impaired cognitive function and a complete failure to reorganize neural circuits.

A key conceptual insight of the current work is the requirement for proper control of synaptic ApoER2 proteins levels for neuronal plasticity and learning. Since loss of ApoER2 impairs LTP, and since ApoER2 mediates Reelin signaling to enhance learning and memory in mice (*Bosch et al., 2016*; *Niu et al., 2008*; *Rogers et al., 2011*), it has been suggested that increased ApoER2 expression would be beneficial. On the contrary, we have shown here that the constitutive ApoER2 expression caused by IDOL deficiency is detrimental to proper dendritic spine morphogenesis and the plasticity of neural circuits. Furthermore, we found that ApoER2 protein expression in mature neurons is not constant, but is acutely regulated at the post-translational level in response to neuronal activity and the induction of LTP. IDOL thus provides a mechanism to actively control ApoER2 levels in order to facilitate the development of new spines and synapses.

Our observation that IDOL bidirectionally regulate spine density through ApoER2 is consistent with previous studies showing that spine density is decreased in ApoER2 KO mice and increased in the setting of ApoER2 overexpression (*Dumanis et al., 2011*; *Wasser et al., 2014*). *Lrp8* lacking exon 16, which encodes the O-linked sugar (OLS) domain, increased the stability of ApoER2 in neurons and was associated with higher spine density (*Wasser et al., 2014*). Interestingly, while mild increases in ApoER2 levels in response to exon 16 deletion led to enhanced LTP (*Wasser et al., 2014*), we showed constitutive high level of ApoER2 expression in IDOL-deficient mice severely impaired LTP. To reconcile these observations, we propose that the dynamic regulation of ApoER2 levels, rather than its absolute abundance, is critical for the functional plasticity of spine synapses. ApoER2 lacking the OLS domain is likely still regulated by IDOL in response to neuronal activity, although its dynamics will be based on the higher ApoER2 level. On the contrary, loss of IDOL completely abolished activity-dependent regulation of ApoER2 levels, which consequently led to impaired structural and functional plasticity of dendritic spines.

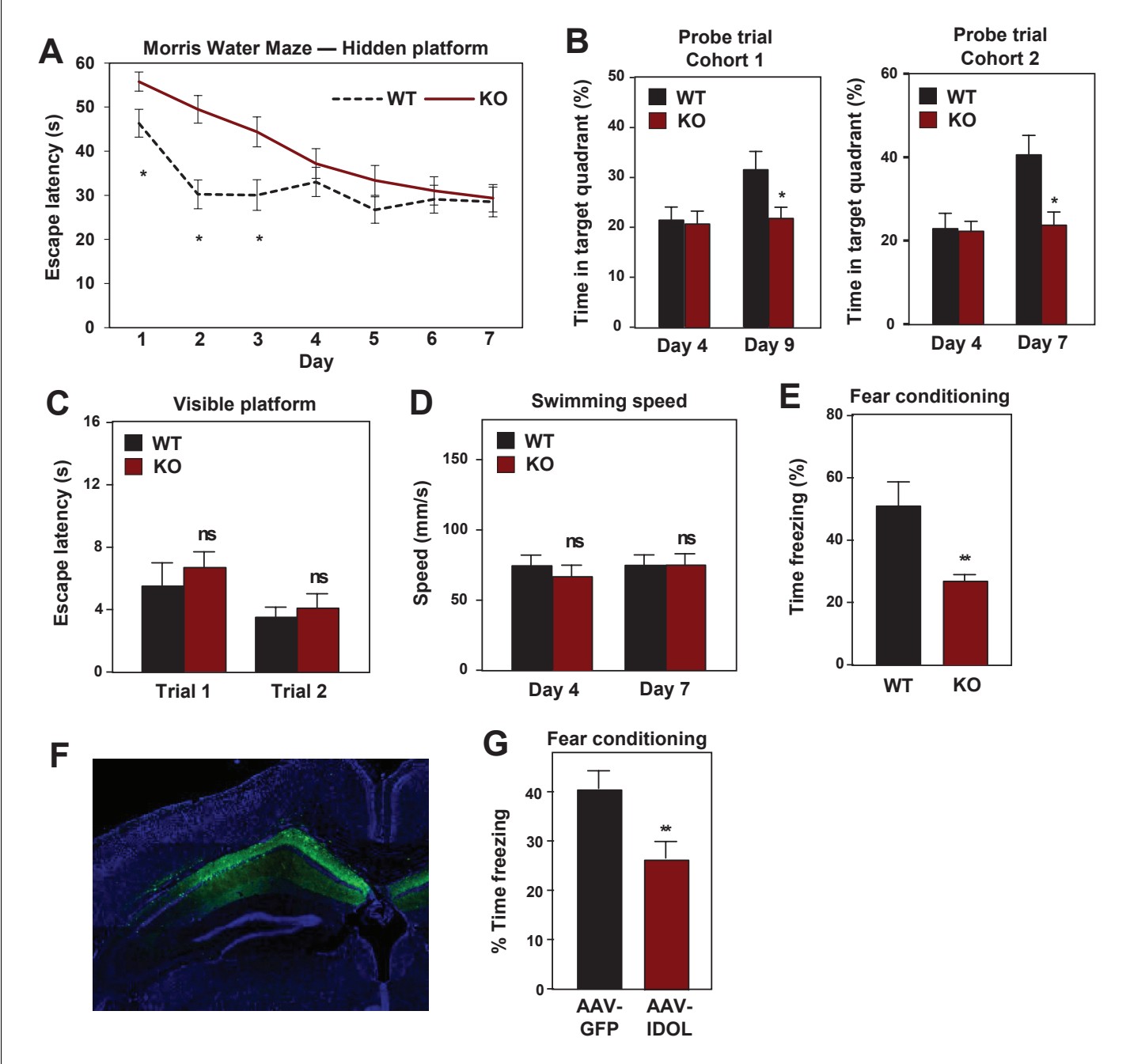

**Figure 8.** IDOL deficiency impairs hippocampal-dependent learning and memory. (**A**) Escape latency to find the hidden platform during training trials of male adult (≥8 month-old) WT and IDOL-deficient mice in the Morris water maze (n = 9/group). *p<0.05 by One-way ANOVA with repeated measures. (**B**) Time spent in the target quadrant searching for the hidden platform after the fourth (first probe trial), seventh or ninth (second probe trial) training trials. (n = 8 for the first cohort; n = 9–10 for the second cohort). *p<0.05 by Student's t test. (**C**) Escape latency to find the visible platform of male adult (≥8 month old) wild type and IDOL-deficient mice in the Morris Water Maze (n = 9–10/group). ns, not significant. (**D**) Swimming speed of male adult (≥8 months old) WT and IDOL-deficient mice. (**E**) Average freezing behavior over 5 min, 24 hr after training, in the same context was measured in female adult (≥8 month old) WT and IDOL-deficient mice (n = 8–10/group). **p<0.01 by Student's t test. (**F**) Representative image of GFP expression in hippocampus two months after stereotaxic vector delivery (AAV-Syn-GFP or AAV-Syn-GFP-IDOL) into the hippocampus of adult mice. (**G**) Contextual memory consolidation was measured by freezing frequency at 24 hr after training in mice injected with AAV-Syn-GFP or AAV-Syn-GFP-IDOL (12 weeks old, n = 10/group). **p<0.01 by Student's t test.

DOI: https://doi.org/10.7554/eLife.29178.016

Our study also suggests that Rac signaling may be involved in the regulation of neuronal structural plasticity by ApoER2. Considerable evidence suggests that spatial and temporal coordination of actin dynamics are prerequisite for synapse formation and plasticity (*Dillon and Goda, 2005*). The Rho small GTPases (particularly RhoA, Rac1, Cdc42) are regulators of the actin cytoskeleton that have a profound influence on spine morphogenesis (*Luo, 2002*; *Newey et al., 2005*). One Rac-GEF implicated in the activity-dependent remodeling of dendritic arbors is Tiam1, which is responsible for the coupling between NMDAR and Rac1 (*Tolias et al., 2005*). We found that IDOL overexpression reduces, while IDOL knockout increases, Rac1 activity in neurons. Loss of IDOL impairs the coupling between glutamate receptor activation and Rac1, increasing basal Rac1 activity and blocking the activation of Rac1 by neuronal activity. The bidirectional regulation of Rac1 activity by IDOL aligns closely with IDOL's impact on spine morphogenesis.

We speculate that ApoER2 may be component of a 'signaling hub' modulating Rac1 activity and actin remodeling of dendritic spines. The JNK-interacting protein (JIP) was originally identified as a putative scaffold that binds components of the JNK signaling pathway (*Whitmarsh et al., 1998*). Recent studies have shown that JIP is localized at post-synaptic densities (PSD) in neurons, and is indispensable for the normal function of NMDA receptors (*Kennedy et al., 2007*; *Pellet et al., 2000*). Our data suggest that postsynaptic JIP (*Stockinger et al., 2000*; *Verhey et al., 2001*) and Tiam1 (*Buchsbaum et al., 2002*) could be involved in mediating the effects of IDOL-ApoER2 axis on Rac1 activity. However, since JIP1 is also an important regulator of axonal development and growth (*Dajas-Bailador et al., 2008*), distinguishing the pre- and post-synaptic functions of JIP1 is technically challenging. Complete elucidation of the molecular components and signaling pathways mediating the effects of IDOL on the morphogenesis and remodeling of dendritic spines will require further investigation.

We aware of the limitations of artificial overexpression systems, and we acknowledge the formal possibility that high-level exogenous expression of IDOL in neurons may have off-target or dominant-negative effects. However, our demonstration that knocking down ApoER2 expression reversed the effects of IDOL expression strongly suggests that these effects are specific. Furthermore, our conclusions from overexpression systems are bolstered by corresponding loss-of-function experiments in genetically modified cells and animals.

What is the benefit of IDOL-dependent ApoER2 regulation for neuronal physiology? IDOL is highly conserved from humans to *drosophila melanogaster*, most likely due to the functions elucidated in this study. We speculate that the IDOL-ApoER2 pathway helps to balance the mobility and stability of dendritic spines, and modulates the speed of spine maturation and synapse formation. High ApoER2 levels during the early postnatal period may serve to keep spines immature and motile. This would be advantageous, because spine motility permits the sampling of large numbers of potential presynaptic partners before connections are stabilized via activity-dependent mechanisms (*Jontes and Smith, 2000*; *Portera-Cailliau et al., 2003*). We further hypothesize that the IDOL-ApoER2 pathway may function as a negative feedback mechanism similar to homeostatic scaling for the maintenance of neuronal network stability. Hebbian plasticity, including LTP and LTD, provoke positive feedback responses and are potentially prone to instability (*Turrigiano, 2008*; *Vitureira and Goda, 2013*). In mature neurons, IDOL expression is suppressed by glutamate receptor activity, leading to the stabilization of ApoER2. Once it passes a certain threshold, this increased ApoER2 expression would be predicted to uncouple activity-induced Rac1 induction and actin polymerization, thus preventing the further enhancement of LTP. The impact of the IDOL-ApoER2 pathway on different forms of synaptic plasticity is an important topic for further investigation. It will also be of great interest to explore the involvement of the IDOL-ApoER2 pathway in the cognitive deficits associated with aging and neurodegenerative disorders.

## Materials and methods

### Animals

*Mylip*$^{-/-}$ and *Mylip*$^{flox/flox}$ mice (backcrossed to C57BL/6 background for nine generations) have been described previously (*Hong et al., 2014*). Wild type littermate or F2 generation derived from F1 intercrosses were used as controls in all animal studies. All mouse experiments were approved

and performed under the guidelines of the Animal Care and Research Advisory Committees at the University of California, Los Angeles (UCLA).

## Antibodies and reagents

Primary antibodies used in this study are: Anti-MAP2 (Cat. 1100, PhosphoSolutions), Anti-PSD95 (7E3-1B8, ThermoFisher), Anti-NMDAR1 (AB9864R, Millipore), Anti-transferrin receptor (TfR) (13–6800, Invitrogen), Anti-NMDAR2A (AB1555P, Chemicon), Anti-NMDAR2B (AB1557P, Chemicon), Anti-p44/42 MAPK (Erk1/2)(9102, Cell Signaling), Anti-Phospho-p44/42 MAPK (Erk1/2)(9106, Cell Signaling), Anti-JIP-1 (B-7)(sc-25267, Santa Cruz), Anti-GluR2 (MAB397, Millipore), Anti-GluR1-NT (N-terminus)(MAB2263, Millipore), Anti-ApoER2 (ab108208, Abcam), Anti-RhoA (ARH04, Cytoskeleton), Anti-Rac1 (ARC03, Cytoskeleton), Anti-LDLR (10007665, Cayman). Anti-VLDLR antibody was a gift of Dr. Joachim Herz, University of Texas Southwestern Medical Center. All secondary antibodies were purchase from ThermoFisher or Jackson Immoresearch. Bicuculline and tetrodotoxin citrate were purchased from Cayman Chemicals. Other chemicals were from Sigma-Aldrich.

The AAV-CamKII-GFP, AAV-CamKII-CRE-GFP, AAV-hSyn-GFP vectors werepurchased from the UNC Vector core. AAV-hSyn-IDOL and AAV-hSyn-IDOL (C387A) vectors were made by th Custom Vector Production service at the UNC Vector Core. High titer lentivirus expressing Cherry, ApoER2 (WT), ApoER2 (F894A), VLDLR (WT), VLDLR (F832A) were generated following the protocol described (Han et al., 2009). GTP-bound (active) Rac1 and RhoA from neuronal cultures was assayed using Rac1(Cat. # BK035) and RhoA (Cat. # BK036) Pull-down Activation Assay Biochem Kits from Cytoskeleton, Inc.

## Primary neuron culture

Primary hippocampal neurons were cultured according published protocols with minor modification (Beaudoin et al., 2012; Kaech and Banker, 2006). Briefly, hippocampi from P0 pups of C57BL/6 or $Mylip^{flox/flox}$ mice were isolated, digested, plated onto poly-L-lysine coated tissue culture dishes or glass coverslips and grown in glia-conditioned Neurobasal A media (Invitrogen) supplemented with 2% B27, 2 mM Glutamax, 50 U/ml penicillin, 50 mg/ml streptomycin. We replaced one third of the fresh medium weekly. Neurons were transfected using Lipofectamine 2000 (Invitrogen). The primary cortical neuron culture protocol was kindly provided by Dr. Richard L. Huganir from Johns Hopkins University. Briefly, cortex from P0 pups was isolated, digested, and plated onto poly-Llysine-coated dishes in Neurobasal A complete growth medium supplemented with 5% fetal bovine serum (FBS). Neurons were switched to 1% FBS Neurobasal medium (conditioned overnight in primary glia cell culture) 24 hr post-seeding and fed twice a week.

## Surface biotinylation

Neurons were rinsed with ice-cold DPBS containing $CaCl2$ and $MgCl2$ (pH 7.4) (ThermoFisher, 14040) once, incubated in DPBS containing 0.5 mg/ml Sulfo-NHS-SS-biotin (Thermo Scientific, 30 min, 4°C), then rinsed once with DPBS and unreacted biotinylation reagent was quenched in DPBS containing 20 mM glycine (twice, 5 min each, 4°C). Cells were then lysed in RIPA lysis buffer (50 mM tris-HCl (pH 7.4), 150 mM NaCl, 0.25% deoxycholic acid, 1% NP-40, 0.1% SDS, 1 mM EDTA) and protease inhibitor cocktail (Roche). Protein concentration of each lysate was quantified using BCA Protein Assay (ThermoFisher). Equal amounts of protein were incubated overnight with Neutr-Avidin coupled agarose beads (ThermoFisher). Beads were washed three times with ice-cold RIPA buffer, and biotinylated proteins were eluted with 2X LDS sample buffer. Cell-surface or total proteins were then separated on NuPAGE bis-tris gels (Invitrogen) and analyzed by western blotting.

## Immunocytochemistry and Golgi staining

Immunostaining was done as described previously (Beaudoin et al., 2012). Briefly, neurons were fixed for 10 min at room temperature in PBS containing 4% paraformaldehyde (PFA) and 4% sucrose, and permeabilized with 0.1% Triton X-100 in PBS. Fixed neurons were then blocked with 10% goat serum in PBS for 60 min, and incubated with primary antibodies in PBS with 1% goat serum overnight at 4°C. Neurons were rinsed, incubated with fluorescently-labeled secondary antibodies, and mounted onto glass slides. Images were taken with a 510 laser-scanning confocal microscope (Zeiss, Germany). The FD Rapid GolgiStain Kit (FD NeuroTechnologies, PK401) was used to

impregnate neurons according to the manual. For each mouse, 100X image of the apical dendrite at CA1 region were obtained. 3 to 4 neurons were analyzed for each animal.

Glutamate receptor internalization was analyzed as described (Kim et al., 2007). Briefly, surface GluR2 receptors were 'live'-labeled with mouse anti-GluR2 N-terminal antibody (MAB397, Chemicon, 10 μg/ml) in conditioned culture medium. After washing in pre-warmed Neurobasal, neurons were either returned to conditioned medium (for control incubation) or stimulated with 50 μM NMDA for 10 min. Neurons were fixed and surface-remaining receptors were visualized with Alexa 647-conjugated secondary antibody. Internalized GluR2 receptors were detected with Alexa 568-conjugated secondary antibody.

## Fractionation of brain tissue

Brain tissues were homogenized in ice-cold HE buffer (10 mM HEPES, 1 mM EDTA, 0.5 mM DTT, protease inhibitor, PH 7.4) for 10 to 15 s/strike using a tissue homogenizer 600 rpm 10 strikes on ice. Centrifuge the remaining homogenate for 10 min at 800 g, 4°C. Supernatant (S1) was separated from the pellet (P1) by decanting or using a transfer pipet. S1 supernatant was then centrifuged for 15 min at 10,000 g, 4°C. The pellet was resuspended in 10 X volume HE buffer containing 320 mM sucrose and then respun at 10,000 x g for another 15 min to yield the washed crude synaptosomal fraction (P2).

## Whole-cell patch recordings

Whole-cell patch recordings were obtained from the soma of 19–21 DIV cultured hippocampal neurons with a Multiclamp 700B amplifier using 2.5–3.5 MΩ patch pipets. The patch pipet solution contained (in mM): 130 CsMeSO$_4$, 10 TEA-Cl, 10 HEPES, 0.5 EGTA, 10 KCl, 2 MgSO$_4$, 4 tris-ATP, 0.3 tris-GTP, adjusted to pH 7.3 with CsOH. The bath solution contained (in mM): 140 NaCl, 10 HEPES, 5 KCl, 3 CaCl$_2$, 1 MgCl$_2$, 20 glucose, 0.001 tetrodotoxin, 0.1 picrotoxin, 0.001 strychnine adjusted to pH 7.4 with NaOH. Recordings were performed at room temperature (22–25°C), and each coverslip was held in the recording chamber for less than one hour. Neurons were held at −70 mV, and traces were low-pass filtered at 2 kHz and digitized at 10 kHz using Axon Instruments Digitizer and pClamp software. To confirm transduction for each neuron recorded, DIC and florescent images were acquired, following each recording, using a Zeiss Axiovert 200 equipped with a Zeiss Axiocam MR camera and Axiovision acquisition software. Recordings were considered for analysis if the series resistance was less than 15 MΩ and if at least 50 mEPSCs were collected. Threshold detection and analysis of mEPSCs was performed using pClamp software. Negative-going events with amplitudes larger than –10 pA and lasting longer than 0.5 ms were automatically detected, and any non-mEPSC events were manually rejected. Student's t-test will be used for the statistical comparisons between genotypes. cLTP induction in cultured hippocampal neurons.

LTP was induced in cultured hippocampal neurons through activation of NMDA receptors with application of 200 μM glycine (Lu et al., 2001). The cells were first treated for 10 min at room temperature with bath solution containing (in mM): 152 NaCl, 10 HEPES, 4 KCl, 2 CaCl$_2$, 10 glucose, 0.001 tetrodotoxin, 0.1 picrotoxin, 0.001 strychnine adjusted to pH 7.4 with NaOH. The cells were then treated with 200 μM glycine in the 0 Mg$^{2+}$ solution at room temperature for 5 min. After this time, the glycine treatment was replaced with the recording bath solution described above for 30 min at room temperature before the coverslip was transferred to the recording chamber for patch recordings.

## In vitro electrophysiology

Adult c57/Bl6J mice (2–3 month old) were deeply anaesthetized with Isoflurane (Isothesia, Henry-Schein, USA) before decapitation. Acute coronal slices (400 μm) through the dorsal hippocampus were cut with a vibratome (VT1000s; Leica Microsystems, Germany) in ice-cold cutting artificial cerebrospinal solution (aCSF) composed of (in mM): 130 NaCl, 3.5 KCl, 1 NaH2PO4, 24 NaHCO3, 1 CaCl2, 3 MgSO4, and 10 D-glucose (all from Sigma, Germany) saturated with 95% O2% and 5% CO2. The slices were immediately transferred to a submerged holding chamber and incubated at ~33°C for 30 min and then at room temperature (RT) for ≥1 hr in the solution used for recording, similar to cutting aCSF, except that it contained 3 mM CaCl2 and 1.5 mM MgSO4 for LTP experiments or 1.5 mM MgSO4 and 4 mM CaCl2 for LTD experiments. All recordings were done in a

submersion recording chamber perfused with aCSF (T = 31 ± 0.5°C) at a flow rate of 2–2.5 ml × min− 1. A bipolar concentric stainless steel electrode (FHC, USA) was placed in the stratum radiatum of the CA1 region of the hippocampus to allow orthodromic stimulation (constant current, 0.2 ms pulses delivered at 0.05 Hz) of the Schaffer collateral/commissural fiber synapses onto CA1 pyramidal cells. Delivery of stimulation via the Digidata 1322A interface (Molecular Devices, USA) was controlled by pClamp 10.4 software (Molecular Devices, USA) and a stimulus isolator (model: A385; WPI, USA). The stimulus intensity was adjusted between 30 and 110 µA to evoke 60% of maximum response, as determined from the input-output curve and the resulting synaptic potentials – field excitatory postsynaptic potentials (fEPSPs). fEPSPs were recorded using an aCSF-filled glass microelectrode (4.5–7 MΩ) placed in stratum radiatum. The recordings were performed with a Multiclamp 700B amplifier (Molecular Devices, USA) and fEPSPs were digitized (Digidata 1322A interface; Molecular Devices, USA), acquired at a sampling rate of 10 kHz, saved to a PC and analyzed off-line with Clampfit 10.4 (Molecular Devices, USA) and OriginPro 9.0 software (OriginLab Corporation, USA). The fEPSPs were monitored for at least 30 min until the amplitudes were generally stable.

Input-Output (IO) curves were established by plotting the fEPSP slope against various intensities of the test pulse, ranging from 0 µA to 110 µA in 10 µA steps. LTP of the Schaffer collateral-CA1 synaptic response was induced by high-frequency stimulation (HFS) consisting of two trains of 100 pulses at 100 Hz (intertrain interval of 10 s) at 100% intensity of the test stimulus. The evoked fEPSPs were recorded for at least 60 min after the HFS. LTD was elicited using a paired-pulse 1 Hz protocol (PP-1Hz) consisting of paired-pulses (50 ms interstimulus interval) repeated at 1 Hz for 15 min, for a total of 900 paired-pulses. The average of fEPSP slopes recorded between 55 and 60 min after HFS or PP-1Hz were used for statistical comparisons. Different slices from the same animal were sometimes used for several experiments, including LTP, LTD or paired-pulse facilitation, but each slice was subjected to only one particular test.

The fEPSP slopes were expressed as a percentage of the 10 min baseline value before the HFS/PP-1Hz. The Mann–Whitney U test was chosen for statistical comparisons of the LTP/LTD data using Origin Pro 9.0 software (OriginLab Corporation, USA). A p value < 0.01 was considered significant.

## Behavior tests
### Morris water maze test
The maze is a stainless-steel circular tanks with 200 cm diameter. The tank is filled water dyed with white, liquid tempera paint to make the water opaque. A platform is placed in one of the quadrants of the pool and submerged 1 cm below the surface. During the trial, mice were placed in the desired start position facing the tank wall, and released into the water at water-level. If a mouse failed to find the platform within 60 s, it was then placed on the platform for 15 s before being removed. Mice were then place at a new start location and the trial repeated for 4 times/day. On days 4, 7 or 9, the probe test (the mouse was allowed to free swim for 60 s with the submerged platform removed) was conducted. Percent time spent in the goal quadrant was calculated and analyzed for each mouse. The experimenter was blind to the genotype of the animals. Behavioral data from the training period were analyzed using repeated measures ANOVA. Data from the probe test were analyzed using one-way ANOVA.

### Fear conditioning
Mice were placed in a shock chamber (Med Associates Inc.) on day 1 and allowed to explore the environment freely for 2 min, followed by a 2 s, 0.5-mA foot-shock. Mice were placed back in their home cage 30 s after foot shock. On day 2, mice were placed in the training context for 5 min, and the level of freezing was recorded using FreezeFrame program and analyzed. One-way ANOVA was used to analyze percent freezing scores of the contextual freezing.

## Optical intrinsic signal imaging
Chronic glass-covered cranial windows were implanted as recently described (*Holtmaat et al., 2009*; *Mostany and Portera-Cailliau, 2008*). Briefly, mice were anesthetized with isoflurane (1.5% via nose cone) and placed in a stereotaxic frame over a warm water re-circulating blanket. Dexamethasone (0.2 mg/kg; Baxter Healthcare Corp.) and carprofen (5 mg/kg; Pfizer) were administered subcutaneously to reduce brain edema and local tissue inflammation. A 4 mm craniotomy was

performed with a pneumatic dental drill. The center of the craniotomy was placed of over the left hemisphere, 2.5 mm lateral to the midline and 2 mm caudal to Bregma. A sterile 5 mm glass cover slip (#1; Electron Microscopy Sciences) was gently laid over the dura mater and glued to the skull with cyanoacrylate-based glue. Dental acrylic was then applied throughout the skull surface. A titanium bar ($0.125 \times 0.375 \times 0.05$ inch) was embedded in the dental acrylic to secure the mouse on to the stage for imaging.

Optical intrinsic signal (OIS) imaging of the hindpaw (HP) and the D2 whisker (D2W) sensory receptive fields was done at 4 different intervals before and after whisker removal: baseline control (average of 2 time points), +1 week, +2 weeks and +3 weeks. OIS imaging was performed through the cranial window on mice under light anesthesia with 0.5–0.75% isoflurane and a single dose of chlorprothixene (6 mg/kg, i.p., Sigma-Aldrich). The cortical surface was illuminated by green (535 nm) and red (630 nm) sets of light-emitting diodes (LEDs) mounted around a 'front-to-front' tandem arrangement of objective lenses (135 mm and 50 mm focal lengths, Nikon). The green LEDs were used to visualize the superficial vasculature and the red LEDs were used for IOS imaging. The microscope was focused to ~350 µm below the cortical surface. Imaging was performed at 10 frames per second using a fast camera (Pantera 1M60, Dalsa), frame grabber (64 Xcelera-CL PX4, Dalsa) and custom routines written in MATLAB (provided in Source Code Files 1 to 6). Each session consisted on 30 trials, taken 20 s apart, of mechanical stimulation for 1.5 s (100 Hz) using a glass micropipette coupled to a piezo bender actuator (Physik Instrumente). Frames 0.9 s before onset of stimulation (baseline) and 1.5 s after stimulation (response) were collected. Frames were binned 3 times temporally and $2 \times 2$ spatially. Stimulated cortical areas were identified by dividing the response signal by the averaged baseline signal (DR/R) for every trial and then summing all trials. Response maps were then thresholded at 50% of maximum response to get the responsive cortical areas for D2W and HP. For figures, we aligned the response maps for D2W and HP stimulation within and across animals for all time points with the help of the corresponding photomicrographs of the superficial vasculature. To generate the final image, we merged and color-coded the responses for D2W and HP (green and red, respectively) for every time point into a single RGB image in Adobe Photoshop (Adobe Systems Inc.).

Cortical sensory representation map sizes were defined by ImageJ software (http://rsbweb.nih.gov/ij/; National Institutes of Health, Bethesda, MD). A two-way ANOVA, followed by Bonferroni's multiple comparison test was used to compute statistical differences between WT and IDOL-deficient groups in Origin Pro 9.0 software (OriginLab Corporation, USA). All data are presented as the mean ±standard error of the mean. Significance was set at $p < 0.01$.

## Acknowledgements

We thank Thomas O'Dell for help in slice electrophysiology experiments, Andrew Poulos, Jesse Cushman, Yong-Seok Lee and Alcino Silva for help with behavioral studies, Joachim Herz for sharing antibodies to ApoER2 and VLDLR, Richard Huganir for the cortical neuron culture protocol, Victoria Ho for help in primary neuron culture, and Jon Salazar for genotyping and mouse care. This research was supported by National Institutes of Health Grants HL066088 and HL090553 (to PT), 4R01NS076942 (to CP-C), and R01MH077022 (to KCM).

## Additional information

### Competing interests

Peter Tontonoz: Reviewing editor, *eLife*. The other authors declare that no competing interests exist.

### Funding

| Funder | Grant reference number | Author |
| --- | --- | --- |
| National Institutes of Health | R01MH077022 | Kelsey C Martin |
| National Institutes of Health | 4R01NS076942 | Carlos Portera-Cailliau |

| Howard Hughes Medical Institute | | Peter Tontonoz |
| National Institutes of Health | HL066088 | Peter Tontonoz |
| National Institutes of Health | HL090553 | Peter Tontonoz |

The funders had no role in study design, data collection and interpretation, or the decision to submit the work for publication.

## Author contributions

Jie Gao, Formal analysis, Supervision, Funding acquisition, Writing—original draft, Project administration, Writing—review and editing; Mate Marosi, Conceptualization, Data curation, Formal analysis, Writing—original draft, Writing—review and editing; Jinkuk Choi, Jennifer M Achiro, Sangmok Kim, Data curation, Formal analysis, Methodology; Sandy Li, Klara Otis, Kelsey C Martin, Data curation, Formal analysis; Carlos Portera-Cailliau, Peter Tontonoz, Formal analysis, Supervision, Writing—review and editing

## Author ORCIDs

Jennifer M Achiro (iD) https://orcid.org/0000-0002-3978-1647
Peter Tontonoz (iD) https://orcid.org/0000-0003-1259-0477

## Ethics

Animal experimentation: This study was performed in strict accordance with the recommendations in the Guide for the Care and Use of Laboratory Animals of the National Institutes of Health. All of the animals were handled according to approved institutional animal care and use committee (IACUC) protocol (#99-131) of the University of California, Los Angeles.

## Decision letter and Author response

Decision letter https://doi.org/10.7554/eLife.29178.024
Author response https://doi.org/10.7554/eLife.29178.025

# Additional files

## Supplementary files

• Source code 1. MATLAB script for CameraSetUp. Activates DALSA camera.
DOI: https://doi.org/10.7554/eLife.29178.017

• Source code 2. MATLAB script for OISpreview. Live camera feed used to find cortical surface.
DOI: https://doi.org/10.7554/eLife.29178.018

• Source code 3. MATLAB script for OIS vasculature. Grabs photo of vasculature on cortical surface.
DOI: https://doi.org/10.7554/eLife.29178.019

• Source code 4. MATLAB script for OISMultitrialPaw. OIS acquisition protocol for whisker/paw stimulation.
DOI: https://doi.org/10.7554/eLife.29178.020

• Source code 5. MATLAB script for OISAnalysis Average. Calculates average data from 30 trials ad subtracts the baseline.
DOI: https://doi.org/10.7554/eLife.29178.021

• Source code 6. MATLAB script for OISOISDisplay. Displays map as MATLAB figure.
DOI: https://doi.org/10.7554/eLife.29178.022

• Transparent reporting form
DOI: https://doi.org/10.7554/eLife.29178.023

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
