## [Decision Letter]

Thank you for submitting your article "The E3 ubiquitin ligase Idol regulates neuronal plasticity through control of synaptic ApoE receptor levels" for consideration by *eLife*. Your article has been favorably evaluated by Gary Westbrook (Senior Editor) and three reviewers, one of whom is a member of our Board of Reviewing Editors. The reviewers have opted to remain anonymous. The reviewers have discussed the reviews with one another and the Reviewing Editor has drafted this decision to help you prepare a revised submission.

Summary:

This study reveals a role for Idol, an E3 ubiquitin ligase, in the regulation of dendritic spines, synaptic plasticity, and associative learning. In support of these conclusions, the authors combine over-expression and knockdown/out of Idol and ApoER2 to show that dynamic regulation of ApoER2 abundance by Idol regulates experience-dependent structural and functional plasticity of neurons.

Essential revisions:

1) The dendritic fragments shown in the manuscript tended to associate with parallel GFP positive filaments, which look like axons. The axon structures intermingled with dendrites. Because both are GFP-positive, it became difficult to recognize isolated dendritic protrusions. Perhaps for this reason, the authors did not provide the results of quantitative analyses for Figure 2 (except Figure 2), although the legend of Figure 2 did mention quantitative analysis. In addition, the presynaptic vGluT1 signals cannot determine the position of postsynaptic spines. To make the data convincing, the authors have to provide better images and use different methods to indicate postsynaptic structures and/or synaptic contact sites. Another concern regarding high infection or transfection efficiency is whether the effect of Idol is purely contributed by the postsynaptic neurons. When both presynaptic and postsynaptic neurons lost or overexpressed Idol, one cannot tell whether the effect is completely contributed from the postsynaptic Idol and its downstream signal.

2) One of the key conclusions is that Idol deletion leads to substantial alterations in the morphology of dendritic spines, as exemplified by decreased spine head width and increased density of (likely immature) spines. This is supported by multiple lines of evidence, including the Golgi staining data in Figure 3. Functionally, these changes are likely to cause decreases in the frequency and/or amplitude of spontaneous excitatory synaptic transmission in mice. Presentation of this kind of functional data would greatly strengthen the manuscript.

3) The authors are strongly encouraged to modify the text in a way to minimize excessive interpretation or discussion, and also substantially tone down the conclusions including the manuscript title. The results shown in the manuscript do not fully support the Idol-ApoER2-Rac1-spine axis in controlling dendritic spine formation and neuronal plasticity.

4) To fit the model, the authors have to first provide comprehensive evidence supporting the essential and sufficient role of ApoER2 in Idol-regulated spine formation, synaptic plasticity and memory. The rescue experiments in dendritic spine formation were only provided in Figure 3. In Figure 4, Figure 5, Figure 6, and 7, the authors investigated the effect of Idol alone, but not ApoER2 rescue. Thus, it is unclear whether ApoER2 is indeed critical for Idol-regulated neural plasticity and memory. As an E3 Ub ligase, Idol is expected to have multiple subtracts. The effects of Idol deletion or overexpression are likely contributed by other Idol subtracts. To conclude the title of the manuscript "The E3 ubiquitin ligase Idol regulates neuronal plasticity through control of synaptic ApoE receptor levels", more rescue experiments in electrophysiology and behavioral analysis have to be included.

5) If Idol acts solely through the downregulation of ApoER2 to alter the spine features, one may expect that in Figure 3, the phenotype of siApoER2/GFP group should be opposite to Idol deletion. Thus, spine density should be decreased and spine head should be larger. However, the authors did not show the results of spine density, which should be included in the manuscript. For the size of spine heads, siApoER2 had actually no effect on GFP control neurons. The data suggest manipulation of ApoER2 alone is not critical for dendritic spine formation. In Idol knockout neurons, ApoER2 knockdown only partially rescued the size of dendritic spines. It also suggests that the effect of Idol is not completely mediated by ApoER2. To explain this controversial observation, the authors brought up a possibility of dynamic regulation of ApoER2. However, there were no data to support the dynamic regulation of ApoER2.

6) The authors tried to link Rac1 and neuronal activity to the Idol-ApoER2 pathway. In Figure 4, Rac1 activity was reduced by Idol overexpression but enhanced by Idol knockout. The data fit the model. However, as the authors were also aware, in Figure 4, they unexpectedly found that in Idol KO neurons, Bic treatment reduced the Rac1 activity to be even lower than that of GFP/Ctrl group and that Idol overexpression enhanced the Rac1 activity in the presence of Bic. These data did not favor the model that NMDAR activation downregulates Idol expression to control Rac1 activity. It suggests that NMDAR likely uses an Idol-independent pathway to inhibit the Rac1 activity.

The part of Idol alone is convincing. However, the causal relation of ApoER2 and Rac1 regulation by Idol in dendritic spine formation and neuronal plasticity is questionable.

7) The Introduction section is well written but some descriptions in the Results and Discussion section lack cogency.

a) Introduction, last paragraph and other, "dynamic regulation of ApoER2 abundance". In the manuscript, there is not any study related to "dynamic regulation".

b) Subsection “Excess Idol activity inhibits spinogenesis by limiting ApoER2 protein”, last paragraph, the authors found that expression of Idol at the late stage did not alter spine density (again, quantitative analysis is required for making this conclusion). In contrast, Idol overexpression at the early stage reduced dendritic spine density. The results suggest the involvement of Idol in the initial stage of spine formation but not maintenance. It is not clear why the authors said that Idol is "not required for synapse formation" in the first paragraph of the Discussion.

c) "the effects of Idol on spinogenesis were preserved in Ldlr-/- neurons (Figure 2—figure supplement 1D)". Figure 2—figure supplement 1D showed only immunoblots of GluR1, GluR2, NR1 and TRF, which cannot be used to indicate "spinogenesis".

d) What is "CR50" in Figure 2—figure supplement 1E?

e) "adding RAP (a high affinity competitive ligand for ApoE receptors) did not abolish the effects of Idol expression (Figure 2—figure supplement 1E). Together, these data suggested that the inhibitory effects of Idol on spinogenesis were likely mediated through ApoER2, and largely independent of the uptake of lipoprotein particles." If Idol expression indeed dramatically reduces ApoER2 expression (Figure 1), there is no or low ApoER2 in Idol expressing cells to respond to RAP. Why do the authors expect to see the effect of RAP?

f) Subsection “Idol-mediated ApoER2 regulation is essential for spine morphogenesis”, first paragraph (Figure 3 and Figure 3—figure supplement 1), again the, images are not clear enough to show the morphology of individual dendritic protrusion (see above point 1). And also the signals of GluT1 alone are not sufficient to conclude synaptic contact sites or excitatory synapses.

g) Figure 3, quantitative analyses of spine density and vGlyT1/Postsynaptic marker double puncta need to be included. The protein levels of ApoER2 in all four groups also need to be examined.

h) "ApoER2 has been reported to interact directly with NR1, but co-immunostaining with vGluT1 revealed that NR1 was appropriately localized on dendritic spines in Idol deficient hippocampal neurons (Figure 4)." In Figure 4, there are only GFP and NR1 signals.

i) "loss of Idol increased protein levels of JIP1 in both total brain lysates and P2 fractions (Figure 4)." Again, the change of JIP1 could be independent from ApoER2. Idol may regulate multiple parallel pathways to control spine formation.

j) "defects in actin remodeling upon NMDAR activation may underlie the failure to induce LTP in the absence of Idol." In Figure 5 and its figure supplements, there is no any data related to actin remodeling.

k) "precise control of neuronal ApoER2 protein levels by Idol is indispensable for synaptic plasticity". The data in the manuscript do not fully support this point.

8) In Figure 4, the authors provide many potential mechanisms that may link Idol deletion with spine alterations. Although the suggested mechanisms are attractive, the causal relationships are weak relative to the data in earlier figures, unless the authors could demonstrate that JIP1 knockdown reverses the spine phenotypes caused by Idol KO. This should be clearly mentioned in Discussion.

9) Although it is possible that both too much and too little Idol or ApoER2 could cause spine abnormalities, in general, interpretation of the results from over-expression of a protein relative to those of gene KO should carefully be interpreted. It is still possible the spine disrupting effects of, for instance, Idol overexpression in Figure 2, although it is E3 activity-dependent, could be a non-physiological dominant negative effect. This should be discussed.

---

## [Author Response]

*Essential revisions:*
*1) The dendritic fragments shown in the manuscript tended to associate with parallel GFP positive filaments, which look like axons. The axon structures intermingled with dendrites. Because both are GFP-positive, it became difficult to recognize isolated dendritic protrusions. Perhaps for this reason, the authors did not provide the results of quantitative analyses for Figure 2 (except Figure 2), although the legend of Figure 2 did mention quantitative analysis. In addition, the presynaptic vGluT1 signals cannot determine the position of postsynaptic spines. To make the data convincing, the authors have to provide better images and use different methods to indicate postsynaptic structures and/or synaptic contact sites. Another concern regarding high infection or transfection efficiency is whether the effect of Idol is purely contributed by the postsynaptic neurons. When both presynaptic and postsynaptic neurons lost or overexpressed Idol, one cannot tell whether the effect is completely contributed from the postsynaptic Idol and its downstream signal.*

We acknowledge the limitation of studying dendritic spine morphology in primary neuron cultures; however, imaging GFP-expressing neurons is still the best available methodology to address the questions we are asking. To address reviewer’s comments regarding the quality of the images and the markers indicating synaptic contact sites, we have added better quality images that distinguish the dendritic protrusions from axons in Figure 2, Figure 3, Figure 3, Figure 4, and Figure 4. We have also included staining for PSD95, a scaffolding protein located in neural postsynaptic densities, to indicate the location of synaptic contact sites in combination with the presynaptic marker vGluT1 in Figure 2, Figure 2, Figure 3, and Figure 4.

Regarding the concern about presynaptic and postsynaptic contributions of Idol, our experimental evidence strongly points to postsynaptic mechanisms. First, the substrate of Idol, ApoER2 is mainly located at postsynaptic terminal of synapses. This is published and we have reconfirmed this in the present study. Second, we show that GFP-tagged, catalytically-inactive Idol fusion protein (GFP-Idol C387A) is distributed in dendritic shafts, and concentrated at synapses in mature hippocampal neurons (Figure 1, and Figure 1—figure supplement 2). Third, we show that overexpression of Idol has major impacts on the levels of postsynaptic proteins including AMPAR, NMDAR1 and PSD95 (Figure 2), and reduces the amplitude of mEPSCs of hippocampal neurons (Figure 3)–both of which indicate a postsynaptic effect. Although we cannot completely rule out the possibility that Idol may affect presynaptic functions, there is no currently no data to support this. In our view it is reasonable to conclude based on all of the evidence outlined above that the majority of the Idol effects we is due to postsynaptic actions.

*2) One of the key conclusions is that Idol deletion leads to substantial alterations in the morphology of dendritic spines, as exemplified by decreased spine head width and increased density of (likely immature) spines. This is supported by multiple lines of evidence, including the Golgi staining data in Figure 3. Functionally, these changes are likely to cause decreases in the frequency and/or amplitude of spontaneous excitatory synaptic transmission in mice. Presentation of this kind of functional data would greatly strengthen the manuscript.*

We assessed the basal synaptic transmission of Schaffer collateral-CA1 synapses by examining the input-output relationship using field recordings of hippocampal slices. We found no difference in the IO curves between WT and *Idol*^–/–^ slices (Figure 6), suggesting that basal synaptic function was not affected by the absence of Idol. In addition, Basal mEPSCs recordings of hippocampal neurons also did not reveal changes in either amplitude or frequency of mEPSCs upon Idol deletion (Figure 6). Both of these datasets suggest that, although loss of Idol affects spine morphogenesis, it does not have a major impact on basal synaptic transmission.

*3) The authors are strongly encouraged to modify the text in a way to minimize excessive interpretation or discussion, and also substantially tone down the conclusions including the manuscript title. The results shown in the manuscript do not fully support the Idol-ApoER2-Rac1-spine axis in controlling dendritic spine formation and neuronal plasticity.*

We have extensively modified the text in response to this concern. We have substantially revised the title, Abstract, and Discussion. The manuscript has also been refocused to emphasize the role of the Idol-ApoER2 axis in regulating spine initiation and morphogenesis, and the associated impairment of spine structural remodeling and synaptic plasticity. We concede that, although an involvement of Rac1 in regulating spine morphology and remodeling is suggested by our data, the underlying signaling pathways remain elusive and thus will require further investigation.

*4) To fit the model, the authors have to first provide comprehensive evidence supporting the essential and sufficient role of ApoER2 in Idol-regulated spine formation, synaptic plasticity and memory. The rescue experiments in dendritic spine formation were only provided in Figure 3. In Figure 4, Figure 5, Figure 6, and 7, the authors investigated the effect of Idol alone, but not ApoER2 rescue. Thus, it is unclear whether ApoER2 is indeed critical for Idol-regulated neural plasticity and memory. As an E3 Ub ligase, Idol is expected to have multiple subtracts. The effects of Idol deletion or overexpression are likely contributed by other Idol subtracts. To conclude the title of the manuscript "The E3 ubiquitin ligase Idol regulates neuronal plasticity through control of synaptic ApoE receptor levels", more rescue experiments in electrophysiology and behavioral analysis have to be included.*

With respect, the reviewer’s statement: “As an E3 Ub ligase, Idol is expected to have multiple subtracts (sic)” is factually incorrect. It is possible that since the reviewer comes from the neuroscience field, he/she may be unfamiliar with our extensive prior work documenting Idol target specificity and elucidating the structural basis for target recognition by Idol (e.g., Calkin et al. PNAS, 2011; Zhang et al., Genes Dev. 2011; Hong et al., JBC 2010). We have already proven that Idol has ONLY three targets: LDLR, VDLR and ApoER2. It does not degrade any other member of the LDLR family, and no other protein in the genome contains the recognition sequence.

In response to the reviewer, we have added new experiments further supporting that, among Idol’s 3 lipoprotein receptor targets, ApoER2 is the major player. Our collective results (including the newly added Figure 3) strongly suggest that ApoER2 mediates the effects of Idol on spine initiation and morphogenesis. First, knockdown of ApoER2 using siRNA markedly reduced synaptogenesis in primary hippocampal neurons; conversely, ApoER2 overexpression resulted in a 2-fold increase in spine density and a ~50% reduction in head width (Figure 4). Both knockdown and overexpression of ApoER2 largely recapitulated the effects of Idol overexpression and deficiency, respectively. Second, expression of an Idol-resistant ApoER2 mutant (F894A) rescued the reduction in filopodia density induced by Idol overexpression (Figure 3), and knocking down ApoER2 in Idol-deficient neurons increased the width of spine heads. Third, our new data show that neurons overexpressing ApoER2 resembled the phenotype of Idol-deficient neurons with respect to Rac1 activity (Figure 5—figure supplement 1). Both showed increased basal Rac1 activity and compromised Rac1 induction upon Bic treatment.

Although we agree it would be desirable to rescue the effects of Idol deficiency by manipulating ApoER2 in vivo, this is extremely challenging and the reviewers did not really offer a constructive and practical suggestion for how this could be accomplished. It is known that the ApoER2 knockout mouse shows defects in neuronal migration, impaired LTP, and learning and memory. Complete loss of ApoER2 impacts dendritic spine initiation, NMDAR signaling, and neuronal positioning as shown by our study and by numerous previous publications. Since the global ApoER2 KO mouse already has so many severe defects, no useful information could be drawn by crossing it to the Idol-knockout mouse. The resulting mice will certainly have defects, but it would be totally unclear how they relate to Idol. Tissue-specific and/or inducible ApoER2 KO mice have not yet been described.

Generating an *inducible and reversible* ApoER2 knockout/knockin mouse model in the background of Idol deficiency could potentially serve as a useful model to study the question; however, such a model will take years to generate and validate and is obviously beyond the scope of an *eLife* revision. We have now revised our manuscript to somewhat soften the conclusions about the in vivo role of ApoER2 vs LDLR and VLDLR in Idol effects.

*5) If Idol acts solely through the downregulation of ApoER2 to alter the spine features, one may expect that in Figure 3, the phenotype of siApoER2/GFP group should be opposite to Idol deletion. Thus, spine density should be decreased and spine head should be larger. However, the authors did not show the results of spine density, which should be included in the manuscript. For the size of spine heads, siApoER2 had actually no effect on GFP control neurons. The data suggest manipulation of ApoER2 alone is not critical for dendritic spine formation. In Idol knockout neurons, ApoER2 knockdown only partially rescued the size of dendritic spines. It also suggests that the effect of Idol is not completely mediated by ApoER2. To explain this controversial observation, the authors brought up a possibility of dynamic regulation of ApoER2. However, there were no data to support the dynamic regulation of ApoER2.*

We are unclear as to the reviewer’s definition of “dynamic”. We show that Idol mediates changes in ApoER2 protein levels in response to neuronal activation and induction of LTP (e.g., Figure X and Figure X). To us, this is a perfectly reasonable use of the term “dynamic”. It does not just determine basal levels of ApoER2, but changes them actively in response to other stimuli. We have limited our use of this term in the revised paper and instead state “Idol is important for mediating acute changes in ApoER2 protein levels in response to…”

Our results (including the newly added Figure 3) strongly suggest that ApoER2 is the main mediator of Idol’s effects on spine initiation and morphogenesis. Please refer to our response to comment 4 for elaboration of the complete supporting evidence. In addition, we showed that ApoER2 protein levels in neurons and in brain is not constant, but is regulated during the early-postnatal period and in response to neuronal activity (Figure 1 and Figure 6) at the post-translational level in Idol-dependent manner.

*6) The authors tried to link Rac1 and neuronal activity to the Idol-ApoER2 pathway. In Figure 4, Rac1 activity was reduced by Idol overexpression but enhanced by Idol knockout. The data fit the model. However, as the authors were also aware, in Figure 4, they unexpectedly found that in Idol KO neurons, Bic treatment reduced the Rac1 activity to be even lower than that of GFP/Ctrl group and that Idol overexpression enhanced the Rac1 activity in the presence of Bic. These data did not favor the model that NMDAR activation downregulates Idol expression to control Rac1 activity. It suggests that NMDAR likely uses an Idol-independent pathway to inhibit the Rac1 activity.*
*The part of Idol alone is convincing. However, the causal relation of ApoER2 and Rac1 regulation by Idol in dendritic spine formation and neuronal plasticity is questionable.*

We have now included new data to show that neurons overexpressing ApoER2 have reduced Rac1 activity upon Bic treatment, resembling the phenotype of Idol KO neurons (Figure 5—figure supplement 1). This result provides further support for a causal relationship between ApoER2 levels and Rac1 activity. However, as pointed out above, we acknowledge that the detailed signaling pathways connecting Idol to Rac1 activity and the structural remodeling of dendritic spines are not yet clear and must be topics for further investigation.

Idol overexpression reduces, while Idol knockout increases, Rac1 activity in neurons. The bidirectional regulation of Rac1 activity by Idol aligns closely with Idol’s impact on spine morphogenesis. Idol overexpression enhanced the Rac1 activity in the presence of NMDAR activation (Bic treatment), consistent with our model that reduced ApoER2 will enhance the coupling between NMDAR and Rac1. It is not clear why Bic treatment reduced Rac1 activity in Idol KO neurons further than in controls. It has been shown that NMDAR activation leads to spatiotemporal activation of Rho GTPases. Rho GTPases work together to coordinate the F-actin organization/remodeling in dendritic spines through their mutual antagonism: for example RhoA antagonizes the activity of Rac1 and CDC42. We speculate that Idol manipulation might specifically decouple NMDAR and Rac1 activation. Reduced Rac1 activity is a possible net outcome due to imbalanced Rho GTPase signaling. Therefore, the reduced the Rac1 activity upon Bic treatment in Idol KO neurons does contradict our working model.

Nevertheless, we have softened our conclusions on this point and instead emphasized other aspects of the Idol-ApoER2 connection in the revised paper.

*7) The Introduction section is well written but some descriptions in the Results and Discussion section lack cogency.*
*a) Introduction, last paragraph and other, "dynamic regulation of ApoER2 abundance". In the manuscript, there is not any study related to "dynamic regulation".*

Again, we do not understand reviewer’s definition of the word “dynamic”. We showed that ApoER2 protein level in neurons and in brain is not constant, but is acutely regulated during the early-postnatal period and in response to neuronal activity (Figure 1 and Figure 6) at the post-translational level in Idol-dependent manner. We have chosen different descriptors for our results in the revised paper to avoid confusion.

*b) Subsection “Excess Idol activity inhibits spinogenesis by limiting ApoER2 protein”, last paragraph, the authors found that expression of Idol at the late stage did not alter spine density (again, quantitative analysis is required for making this conclusion). In contrast, Idol overexpression at the early stage reduced dendritic spine density. The results suggest the involvement of Idol in the initial stage of spine formation but not maintenance. It is not clear why the authors said that Idol is "not required for synapse formation" in the first paragraph of the Discussion.*

We apologize for not making this point more clearly. We meant to state that “Idol is not essential for basal synaptic transmission”, which is supported by the electrophysiology results both in vitro and in vivo. We have revised the sentences accordingly.

*c) "the effects of Idol on spinogenesis were preserved in Ldlr-/- neurons (Figure 2—figure supplement 1D)". Figure 2—figure supplement 1D showed only immunoblots of GluR1, GluR2, NR1 and TRF, which cannot be used to indicate "spinogenesis".*

Our data suggest that forced Idol expression reduced the level of key post-synaptic proteins by inhibiting spinogenesis. And these effects of Idol were preserved in Ldlr-/- neurons. However, we agree with the reviewer that immunoblots of post-synaptic proteins cannot be used to indicate “spinogenesis”, we have modified the sentence to "the effects of Idol on levels of key post-synaptic proteins were preserved in *Ldlr^–/–^* neurons".

*d) What is "CR50" in Figure 2—figure supplement 1E?*

CR50 is a neutralizing antibody for Reelin which binds to ApoER2 and regulates neuronal migration and positioning in the developing brain. We have added an explanatory sentence in the Results section.

*e) "adding RAP (a high affinity competitive ligand for ApoE receptors) did not abolish the effects of Idol expression (Figure 2—figure supplement 1E). Together, these data suggested that the inhibitory effects of Idol on spinogenesis were likely mediated through ApoER2, and largely independent of the uptake of lipoprotein particles." If Idol expression indeed dramatically reduces ApoER2 expression (Figure 1), there is no or low ApoER2 in Idol expressing cells to respond to RAP. Why do the authors expect to see the effect of RAP?*

Because blocking lipoprotein binding could affect WT cells. If the binding of ApoE-containing lipoprotein particles to ApoER2 is required for the effects of Idol, we reasoned that blocking their binding by RAP would mimic the effects of Idol in the WT neurons. Assuming Idol and RAP act on the same pathway, then the combination of both will not have additive effects, and the effects of Idol will be abolished in the presence of RAP. We observed that addition of RAP to primary neuron culture had little effect on the level of key post-synaptic proteins (in WT or IDOL KO), and the effects of Idol remain intact in the presence of RAP. Therefore, we concluded that the effects of Idol on spinogenesis are likely independent of the uptake of lipoprotein particles.

*f) Subsection “Idol-mediated ApoER2 regulation is essential for spine morphogenesis”, first paragraph (Figure 3 and Figure 3—figure supplement 1), again the, images are not clear enough to show the morphology of individual dendritic protrusion (see above point 1). And also the signals of GluT1 alone are not sufficient to conclude synaptic contact sites or excitatory synapses.*

We have now added better quality images to show morphology of the dendritic protrusions (Figure 4, Figure 4). We have also included staining for PSD95 to indicate the location of synaptic contact sites in combination with presynaptic vGluT1 in Figure 2, Figure 2, Figure 3, Figure 4.

*g) Figure 3, quantitative analyses of spine density and vGlyT1/Postsynaptic marker double puncta need to be included. The protein levels of ApoER2 in all four groups also need to be examined.*

Quantitative analysis of spine density, and/or co-staining of pre-synaptic marker vGLuT1 and postsynaptic marker PSD 95 have been included in Figure 1; Figure 2; Figure 3. The protein levels of ApoER2 in all conditions are now shown in Figure 5 and Figure 4—figure supplement 1.

*h) "ApoER2 has been reported to interact directly with NR1, but co-immunostaining with vGluT1 revealed that NR1 was appropriately localized on dendritic spines in Idol deficient hippocampal neurons (Figure 4)." In Figure 4, there are only GFP and NR1 signals.*

We have added vGluT1 staining in Figure 5 (originally Figure 4) to show the co-immunostaining of NR1 and vGluT1.

*i) "loss of Idol increased protein levels of JIP1 in both total brain lysates and P2 fractions (Figure 4)." Again, the change of JIP1 could be independent from ApoER2. Idol may regulate multiple parallel pathways to control spine formation.*

We concede this point. Although our data suggest the potential involvement of postsynaptic JIP and Tiam1 in mediating effects of Idol-ApoER2, the underlying signaling pathways require further investigation. We have added a paragraph to discuss these limitations.

*j) "defects in actin remodeling upon NMDAR activation may underlie the failure to induce LTP in the absence of Idol." In Figure 5 and its figure supplements, there is not any data related to actin remodeling.*

Actin filaments are the major cytoskeletal component of dendritic spines, and actin remodeling is known to dictate the structural arrangement of dendritic spines in response to synaptic activities. We have shown that Idol-deficient neurons failed to undergo morphology changes in response to Glycine treatment (chemical-LTP). Furthermore, Idol manipulation alters Rac1 activity which is known to dictate actin polymerization and spine head enlargement during chemical LTP. In response we have now used the term “spine structural remodeling” instead of actin remodeling.

*k) "precise control of neuronal ApoER2 protein levels by Idol is indispensable for synaptic plasticity". The data in the manuscript do not fully support this point.*

We have toned down our conclusions. We have revised the manuscript to focus on the effects of Idol in regulating spine initiation and morphogenesis, and the associated impairment of spine structural remodeling and synaptic plasticity.

*8) In Figure 4, the authors provide many potential mechanisms that may link Idol deletion with spine alterations. Although the suggested mechanisms are attractive, the causal relationships are weak relative to the data in earlier figures, unless the authors could demonstrate that JIP1 knockdown reverses the spine phenotypes caused by Idol KO. This should be clearly mentioned in Discussion.*

As JIP1 is also an important regulator of axonal development and axonal growth, distinguishing the pre- and post-synaptic function of JIP1 is technically challenging. A discussion regarding the limitations of our study with respect to JIP1 has been added to the revised paper.

*9) Although it is possible that both too much and too little Idol or ApoER2 could cause spine abnormalities, in general, interpretation of the results from over-expression of a protein relative to those of gene KO should carefully be interpreted. It is still possible the spine disrupting effects of, for instance, Idol overexpression in Figure 2, although it is E3 activity-dependent, could be a non-physiological dominant negative effect. This should be discussed.*

We have now added a paragraph to discuss the limitations of overexpression experiments as requested.